# Mature parvalbumin interneuron function in prefrontal cortex requires activity during a postnatal sensitive period

**Sarah E Canetta**[1,2]*[†], **Emma S Holt**[2], **Laura J Benoit**[1,3], **Eric Teboul**[3], **Gabriella M Sahyoun**[2], **R Todd Ogden**[4], **Alexander Z Harris**[1,5]‡, **Christoph Kellendonk**[1,3,6]*‡

[1]Department of Psychiatry, Columbia University Medical Center, New York, United States; [2]Division of Developmental Neuroscience, New York State Psychiatric Institute, New York, United States; [3]Division of Molecular Therapeutics, New York Psychiatric Institute, New York, United States; [4]Department of Biostatistics, Mailman School of Public Health, Columbia University Medical Center, New York, United States; [5]Division of Integrative Neuroscience, New York State Psychiatric Institute, New York, United States; [6]Department of Molecular Pharmacology & Therapeutics, Columbia University Medical Center, New York, United States

**\*For correspondence:**
ses2119@cumc.columbia.edu (SEC);
ck491@cumc.columbia.edu (CK)

[†]Lead contact

[‡]Senior author

**Competing interest:** The authors declare that no competing interests exist.

**Abstract** In their seminal findings, Hubel and Wiesel identified sensitive periods in which experience can exert lasting effects on adult visual cortical functioning and behavior via transient changes in neuronal activity during development. Whether comparable sensitive periods exist for non-sensory cortices, such as the prefrontal cortex, in which alterations in activity determine adult circuit function and behavior is still an active area of research. Here, using mice we demonstrate that inhibition of prefrontal parvalbumin (PV)-expressing interneurons during the juvenile and adolescent period, results in *persistent* impairments in adult prefrontal circuit connectivity, in vivo network function, and behavioral flexibility that can be reversed by targeted activation of PV interneurons in adulthood. In contrast, reversible suppression of PV interneuron activity in adulthood produces no lasting effects. These findings identify an activity-dependent sensitive period for prefrontal circuit maturation and highlight how abnormal PV interneuron activity during development alters adult prefrontal circuit function and cognitive behavior.

## Editor's evaluation

The authors explored the time-dependent effects of inhibition of parvalbumin-expressing interneurons (PV cells) in the mouse prefrontal cortex on task learning and cognition. Overall, the important study provides solid evidence showing that prefrontal cortex PV cell activity during a sensitive period of neurodevelopment affects prefrontal cortex function in the adult mouse. This study progresses our understanding of the development of the prefrontal cortex.

## Introduction

In adults, the prefrontal cortex (PFC) is essential for cognitive processes like working memory and behavioral flexibility, and alterations in prefrontal function are believed to underlie impairments in these behaviors in disorders such as schizophrenia and attentional deficit hyperactivity disorder (*Bolkan et al., 2017*; *Brown and Tait, 2016*; *Millan et al., 2012*). The PFC is a late-developing structure, whose circuitry continues to mature through adolescence into young adulthood. Indeed,

adolescence appears to be a particularly important time for prefrontal maturation as environmental and pharmacological experiences in this period can elicit long-lasting effects on prefrontal function and behavior (*Makinodan et al., 2012*; *Yamamuro et al., 2018*; *Bicks et al., 2020*; *Thomases et al., 2013*; *Labouesse et al., 2017*). This enhanced sensitivity to environmental risk factors is thought to contribute to the common onset of psychiatric disorders during adolescence (*Mayo et al., 2017*; *Andréasson et al., 1987*).

Classic work in the visual system provides an important framework for understanding how transient alterations in environmental experience can lead to persistent effects on adult brain functioning and behavior. In this system, a loss of visual experience during a sensitive period in development, results in activity-dependent remodeling of visual cortical circuitry and long-lasting impairments in visual functioning in adulthood (*Hensch, 2005*). Recent studies in mice demonstrate that cortical activity may play an important role in sculpting PFC development during an early period spanning late prenatal into early neonatal development in humans (*Bitzenhofer et al., 2021*). However, extensive refinement of prefrontal circuitry continues through adolescence, and whether neuronal activity during this later period is required for prefrontal circuit maturation remains an open question (*Chini and Hanganu-Opatz, 2020*).

Within the PFC, parvalbumin (PV)-expressing interneurons regulate the activity of excitatory cortical pyramidal neurons (*Markram et al., 2004*; *Wonders and Anderson, 2006*). Moreover, PV interneurons undergo a protracted period of physiological maturation and integration into cortical circuitry during the juvenile and adolescent period, prior to attaining their mature phenotype (*Yang et al., 2014*; *Miyamae et al., 2017*). The importance of this cell population for cortical development is underscored by the fact that they show histological abnormalities in multiple neurological and psychiatric disorders with a presumed neurodevelopmental origin, most classically schizophrenia (*Lewis, 2014*; *Ruden et al., 2021*). Whether activity of PV interneurons during development is required for proper prefrontal circuit maturation, allowing for optimal cognitive functioning in adulthood, remains unknown.

To address this question, we reversibly inhibited PV interneurons in the medial PFC (mPFC) in the mouse during postnatal development using the DREADD receptor, hM4DG$_i$ (*Roth, 2016*), and studied the behavioral and physiological consequences in adulthood. Targeted regions included the prelimbic, infralimbic, and cingulate cortex chosen because of their shared functional homology with human dorsolateral PFC, which, as previously noted, is implicated in normal cognitive functioning and compromised in psychiatric disorders such as schizophrenia. Using this approach, we found that *reversibly* decreasing mPFC PV interneuron activity during juvenile and adolescent development (postnatal day P14–50), produces *persistent* impairments in adult extradimensional (ED) set-shifting behavior, with corresponding deficits in PV interneuron-pyramidal cell functional connectivity and task-evoked gamma oscillations. This critical deficit in task-evoked gamma oscillations resulted in a corresponding inability to encode correct versus incorrect task outcome. Additionally, this juvenile period is particularly sensitive for prefrontal circuit maturation as comparable inhibition of mPFC PV interneurons in adulthood did not induce long-lasting changes in behavior and electrophysiology. Strikingly, we could rescue the cognitive behavior and task-induced gamma oscillations in developmentally inhibited mice by acutely enhancing mPFC PV interneuron excitability in adulthood, demonstrating that cognitive impairments and prefrontal network function can be rescued even in the context of a developmentally altered brain.

Together, this work provides evidence that PV activity during the juvenile to adolescent time window is required for prefrontal network maturation. These findings highlight how abnormal early activity increases the risk for later maladaptive circuit function and behavior and provide hope that targeted manipulations can reverse developmentally induced cognitive deficits.

## Results
### A chemogenetic system to reversibly inhibit mPFC PV interneurons during development or adulthood

In order to reversibly inhibit mPFC PV interneuron activity during discrete windows in development or adulthood we expressed the DREADD receptor, hM4DG$_i$, specifically in medial prefrontal PV-expressing interneurons from a young age. Postnatal day 10 (P10) pups expressing Cre recombinase

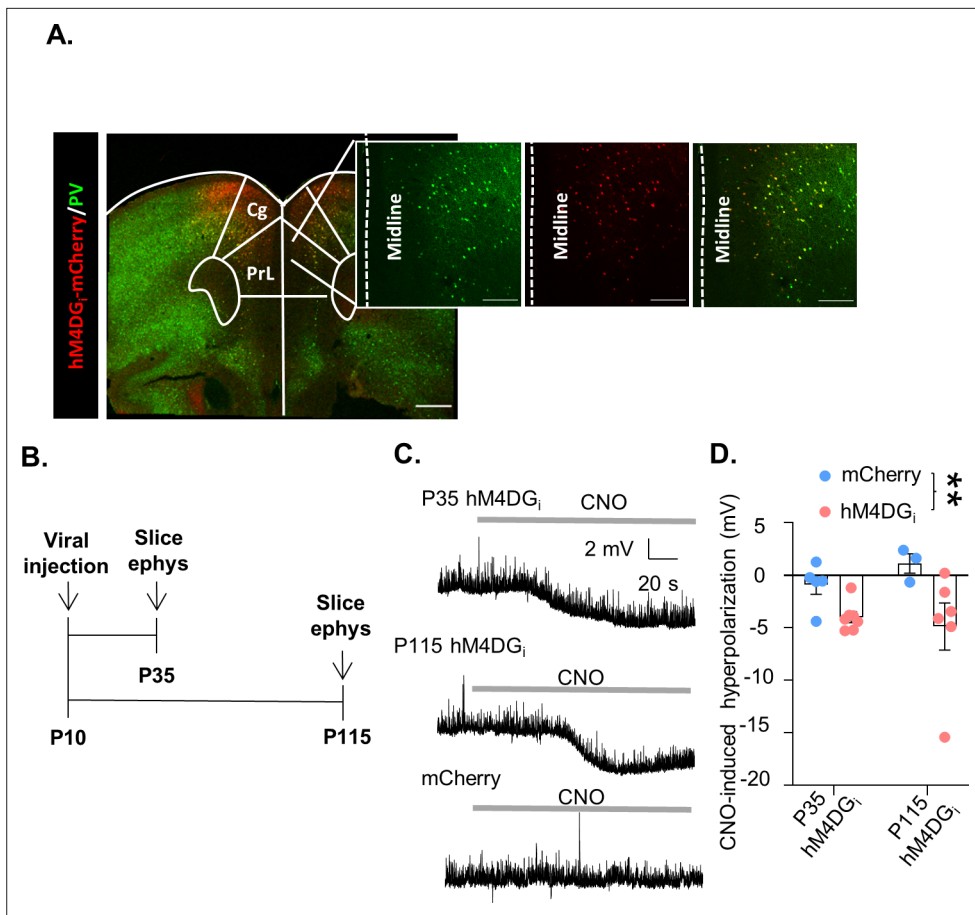

**Figure 1.** A chemogenetic system to reversibly inhibit medial PFC (mPFC) parvalbumin (PV) interneurons during development or adulthood. (**A**) Illustration of hM4DG$_i$-mCherry viral (red) and PV (green) expression in mPFC PV cells (PrL: prelimbic; Cing: cingulate). (**B**) Timeline of experiments for validating the function of hM4DG$_i$. Mice were injected with virus at P10, and whole-cell patch clamp recordings (slice ephys) were made at P35 and P115 from cells expressing hM4DG$_i$-mCherry or mCherry at baseline and in response to bath application of 10 µM clozapine-*n*-oxide (CNO). (**C**) Representative traces illustrating hyperpolarization of the resting membrane potential of hM4DG$_i$-expressing, but not mCherry-expressing, cells following bath application of CNO. (**D**) Quantification of CNO-induced hyperpolarization. CNO-induced hyperpolarization was significantly greater in hM4DGi-expressing cells than in mCherry-expressing cells, regardless of the age that the measurements were made (*n* = 5 P35 and 3 P115 mCherry-expressing cells and 7 P35 and 6 P115 hM4DG$_i$-expressing cells). Dots indicate individual cell responses and bars indicate mean ± standard error of the mean (SEM). Significance evaluated by two-way analysis of variance (ANOVA). Scale bars are 250 µm. **p < 0.01.

The online version of this article includes the following source data and figure supplement(s) for figure 1:

**Source data 1.** Slice electrophysiology data relevant to *Figure 1*.

**Figure supplement 1.** Characterization of viral expression.

**Figure supplement 1—source data 1.** Viral expression data relevant to *Figure 1—figure supplement 1*.

under the PV promoter were injected with an adeno-associated virus (AAV) carrying either a Cre-dependent form of hM4DG$_i$-mCherry or the fluorescent marker, mCherry, into the mPFC. Histological assessment of virus expression verified that hM4DG$_i$-mCherry was restricted to PV-expressing cells predominantly in the prelimbic, infralimbic, and anterior cingulate areas, with some minor spread to the adjacent frontal motor regions of the PFC (*Figure 1*, *Figure 1—figure supplement 1*). Slice electrophysiological recordings made in hM4DG$_i$-mCherry mice at P35 and P115 verified the functional effects of hM4DG$_i$ in PV interneurons (*Figure 1B*, time points chosen to be in the middle of the developmental and adult inhibition windows, respectively). Bath application of 10 µM clozapine-*n*-oxide (CNO) hyperpolarized the resting membrane potential of hM4DG$_i$-expressing cells at both P35

and P115, relative to mCherry-expressing cells. A two-way analysis of variance (ANOVA) looking at the effects of tissue age and virus used on CNO-induced hyperpolarization found a significant effect of virus, but no effect of age or interaction of age and virus (*Figure 1C, D*, mean ± standard error of the mean [SEM]: 0.89 ± 0.94, 1.13 ± 0.91, −3.99 ± 0.52 mV, and −4.88 ± 2.24 mV from *n* = 5 P35 and 3 P115 PV-mCherry cells and *n* = 7 P35 and 6 P115 hM4D cells, respectively, main effect of virus $F(1,17) = 9.069$, p = 0.0079 but no main effect of age $F(1,17) = 0.1391$, p = 0.7138 nor interaction of age and virus $F(1,17) = 0.9260$, p = 0.3494). These results demonstrate that acute CNO administration decreases the excitability of mPFC PV interneurons expressing hM4DG$_i$ both early during development and in adulthood. We also used stereology to quantify the percent of PV cells that were targeted using our viral injection strategy and found that on average 69 ± 7% of PV cells in the prelimbic cortex expressed mCherry and 67 ± 6% expressed hM4DG$_i$-mCherry (mean ± SEM; *n* = 5 mCherry and *n* = 4 hM4DG$_i$-mCherry animals; *Figure 1—figure supplement 1D*).

## Developmental inhibition of mPFC PV interneurons results in persistent alterations in behavior and prefrontal network functioning in adulthood

To test whether reversibly inhibiting the activity of mPFC PV cells during juvenile and adolescent development results in persistent effects on behavior and prefrontal network function, we injected P10 PV-Cre pups with an AAV-expressing hM4DG$_i$ or mCherry and administered CNO between P14 and P50 (subsequently referred to as 'Dev Inhibition' or 'Dev Control', respectively; *Figure 2A*). We chose P14–50 to test a broad time window that encompasses multiple aspects of mPFC PV interneuron maturation (*Miyamae et al., 2017*; *Yang et al., 2014*; *Le Magueresse and Monyer, 2013*). Forty days following the last injection of CNO, when the animals reached adulthood (≥P90) (*Chini and Hanganu-Opatz, 2020*), we initiated testing in an odor- and texture-based attentional set-shifting task (the rodent equivalent of the Wisconsin Card Sorting task), which tests cognitive flexibility. In this task, mice dig into two bowls filled with one of two different textures and scented with one of two different odors (*Figure 2B*). Mice initially learn that odor but not texture predicts reward. During the ED set-shift the mouse has to learn that the rule shifted from the odor dimension to the texture dimension. We and others have shown that the ED shift is dependent on mPFC PV interneurons (*Canetta et al., 2016*; *Cho et al., 2015*; *Goodwill et al., 2018*; *Cho et al., 2020*). We also implanted the mice with electrodes to record neural activity in the mPFC during behavior to assess prefrontal network function.

We found that mice that experienced a reversible inhibition of mPFC PV interneuron activity during development were impaired in ED set-shifting as adults. Although the developmental inhibition did not affect the initial acquisition (IA) of the task (*Figure 2—figure supplement 1A*, Dev Control: mean ± SEM: 13.25 ± 1.19 trials, *n* = 8 mice; Dev Inhibition: 13.2 ± 2.01 trials, *n* = 5 mice; unpaired *t*-test, p = 0.9822), it delayed the acquisition of ED set-shifting portion of the task (*Figure 2C*, Dev Control: mean ± SEM: 11 ± 1.12 trials, *n* = 8 mice; Dev Inhibition: 20.4 ± 2.42 trials, *n* = 5 mice; unpaired *t*-test, p = 0.0021). This impairment was also seen as a significant increase in the total number of errors performed during the ED set-shifting portion of the task (*Figure 2C*, Dev Control: mean ± SEM: 2.25 ± 0.7 errors, *n* = 8 mice; Dev Inhibition: 7.8±1.02, *n* = 5 mice; unpaired *t*-test, p = 0.0007). The increased number of errors after developmental inhibition was seen across both perseverative (*Figure 2—figure supplement 1B*, Dev Control: mean ± SEM: 61.11 ± 15.52%, *n* = 6 mice; Dev Inhibition: 51.39 ± 6.33%, *n* = 5 mice; unpaired *t*-test, p = 0.6042) and random error types (*Figure 2—figure supplement 1C*, Dev Control: mean ± SEM: 38.89 ± 15.53%, *n* = 6 mice; Dev Inhibition: 48.61 ± 6.33%, *n* = 5 mice; unpaired *t*-test, p = 0.6042). Developmental inhibition did not alter the time it took the mice to complete a trial, suggesting their impairment was not due to a decrease in motivation (*Figure 2—figure supplement 1D, E*, IA, Dev Control: mean ± SEM: 39.32 ± 15.98 s, *n* = 8 mice; Dev Inhibition: 26.79 ± 7.53 s, *n* = 5 mice, unpaired *t*-test, p = 0.5690; ED set-shifting, Dev Control: mean ± SEM: 41.76 ± 10.08 s, *n* = 8 mice; Dev Inhibition: 39.95 ± 12.28 s, *n* = 5 mice; unpaired *t*-test, p = 0.9123). Further, developmental inhibition did not affect movement in the open field test (*Figure 2—figure supplement 1F*, Dev Control: mean ± SEM: 8776 ± 580 movements, *n* = 8 mice; Dev Inhibition: 8328 ± 390 movements, *n* = 5 mice; Mann–Whitney, p = 0.8329), indicating no overall motor impairment in these animals.

Intriguingly, we found that in Dev Control animals, prefrontal power in the gamma frequency increased just prior to when the animals made their choice in the task (signified by digging in one of the pots), relative to gamma power recorded in the beginning of the task (here on referred to as

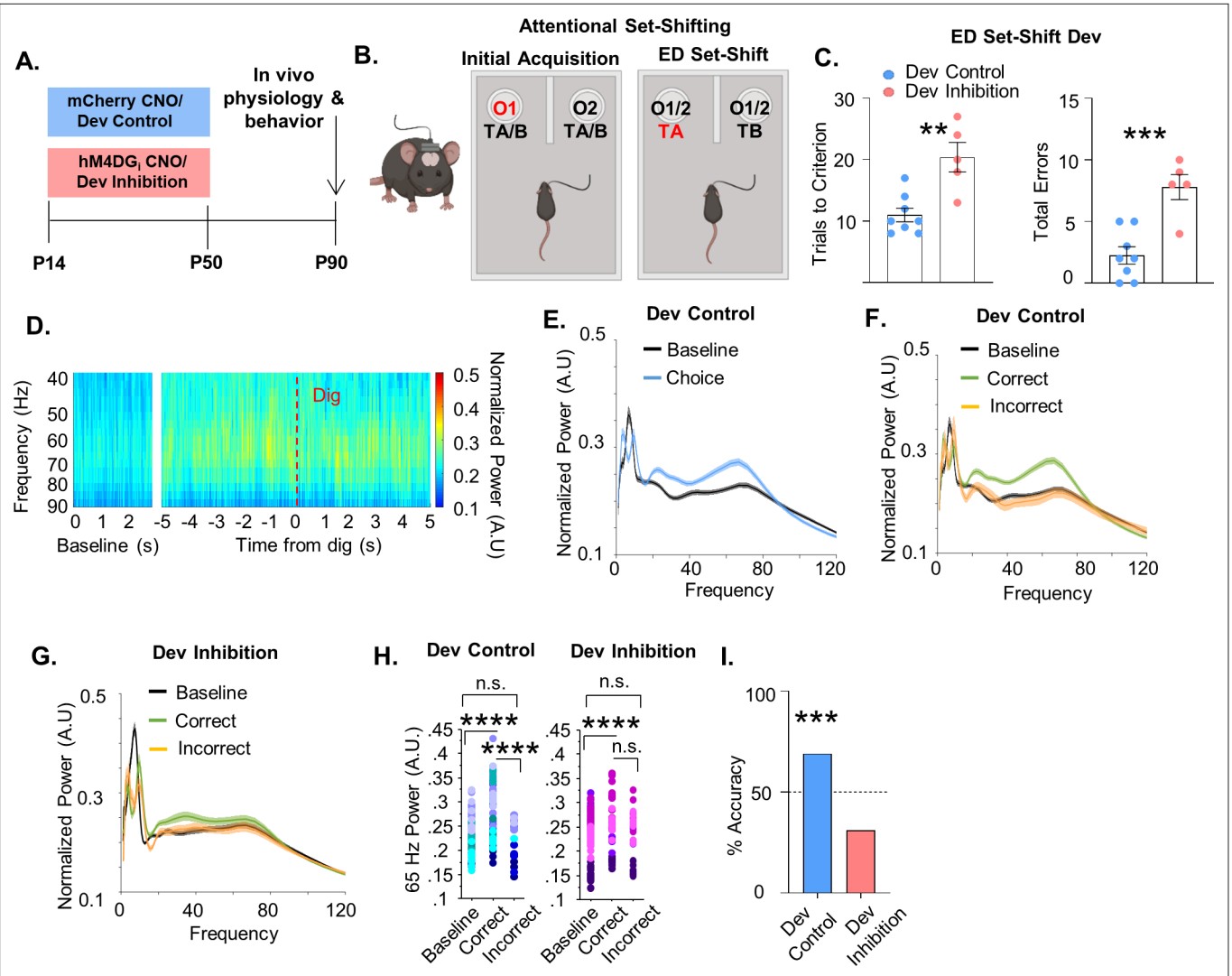

**Figure 2.** Developmental inhibition of medial PFC (mPFC) parvalbumin (PV) interneurons results in persistent alterations in prefrontal network functioning and behavior in adulthood. (**A**) Experimental timeline. Clozapine-n-oxide (CNO) was administered between P14 and P50 to mice-expressing hM4DG$_i$-mCherry or mCherry in mPFC PV interneurons to produce Dev Inhibition or Control mice, respectively. At P90 mice were evaluated in an attentional set-shifting task while local field potentials were simultaneously recorded in the mPFC as illustrated in the schematic in (**B**). (**C**) Dev Inhibition mice ($n$ = 5) take significantly more trials to reach criterion (left) and make more errors (right) than Dev Controls ($n$ = 8). Dots indicate individual animal responses and bars indicate mean ± standard error of the mean (SEM). (**D**) Heat map depicting normalized power (in artificial units, A.U.) as a function of frequency in the first 3 s of the task (baseline) and in a 10-s window centered around when the Dev Control animals make their choice (dig, dotted red line). Gamma frequency activity (here shown between 40 and 90 Hz) increases just prior to choice. (**E**) Normalized power versus frequency in the 3 s preceding choice ('choice', blue line) versus in the first 3 s of the task ('baseline', black line) in Dev Control animals ($n$ = 79 baseline and choice trials from 8 mice). (**F**) Normalized power versus frequency for the choice period prior to correct (green) or incorrect (orange) choices, or in the baseline (black) period for Dev Control ($n$ = 79 baseline/62 correct/17 incorrect trials from 8 mice) or (**G**) Dev Inhibition animals ($n$ = 69 baseline/41 correct/28 incorrect trials from 4 mice). Lines and shading indicate mean ± SEM. (**H**) 65-Hz frequency range power is significantly elevated just prior to correct, but not incorrect, choices relative to baseline for both Dev Control (left, blue dots) and Dev Inhibition (right, pink/purple dots) animals. 65-Hz frequency range power is significantly greater in correct versus incorrect choices in Dev Control, but not Dev Inhibition, mice. (**I**) Choice period 65-Hz frequency range power can predict trial outcome in Dev Control animals (left, blue bar, $n$ = 54 accurately predicted outcomes of 78 tested trials) but not Dev Inhibition animals (right, light red bar, $n$ = 20 accurately predicted outcomes of 64 tested trials). Bar indicates accuracy of model. Dotted line indicates chance level (50%). Significance determined by unpaired $t$-test (**C**) mixed-effects linear regression (**H**), or binomial test (**I**). **$p < 0.01$; ***$p < 0.001$; ****$p < 0.0001$.

The online version of this article includes the following source data and figure supplement(s) for figure 2:

**Source data 1.** Behavioral data relevant to *Figure 2*.

**Source data 2.** 65-Hz frequency range power for dev control mice relevant to *Figure 2*.

**Source data 3.** 65-Hz frequency range power for dev inhibition mice relevant to *Figure 2*.

*Figure 2 continued on next page*

*Figure 2 continued*

**Source data 4.** Power predictor accuracy values relevant to *Figure 2*.

**Figure supplement 1.** Additional behavioral analysis related to the medial PFC (mPFC) parvalbumin (PV) interneuron developmental inhibition.

**Figure supplement 1—source data 1.** Behavioral data relevant to *Figure 2—figure supplement 1*.

**Figure supplement 2.** Additional in vivo electrophysiological analysis related to the selection of the frequency range for analysis.

**Figure supplement 2—source data 1.** Data to select frequency range for LFP analysis relevant to *Figure 2—figure supplement 2*.

**Figure supplement 3.** Accuracy of trial outcome prediction based on the power of different frequencies for Dev Inhibition and Control Mice.

**Figure supplement 3—source data 1.** Data for LFP power predictor accuracy analysis relevant to *Figure 2—figure supplement 3*.

**Figure supplement 4.** Additional analyses of gamma power during the set-shifting task.

**Figure supplement 4—source data 1.** 65-Hz frequency range power for dev inhibition and control animals relevant to *Figure 2—figure supplement 4*.

'baseline'; *Figure 2D, E*). Notably, this increase in task-induced gamma was seen only in trials in which the animal proceeded to make the correct choice and not the incorrect choice (*Figure 2F*). Gamma oscillations can encompass a broad frequency range spanning from 30 to 120 Hz (*Sohal, 2016*). To capture the oscillation frequency most relevant to the behavior, we identified 62–67 Hz (henceforth referred to as 65 Hz) as the frequency range within this 30–120 Hz gamma span where correct trials most frequently showed the maximal difference from incorrect trials (identified using the *max* function in Matlab, *Figure 2—figure supplement 2A*) and that showed the largest magnitude of difference between correct and incorrect trials (*Figure 2—figure supplement 2B*). We confirmed that this was the frequency range that was also maximally different between correct and incorrect trials in a second control group of mCherry animals given CNO in adulthood (see Adult Controls; *Figure 2—figure supplement 2*). Correct trials had significantly higher 65-Hz frequency range power relative to incorrect trials and to the baseline period (*Figure 2H*, 79 baseline/62 correct/17 incorrect trials from 8 Dev Control mice, mixed effects linear regression, correct versus baseline p < 0.0001, incorrect versus baseline, p = 0.462, correct versus incorrect p < 0.0001).

Since 65-Hz frequency range power differed between the choice period of correct and incorrect trials, we determined whether it could predict task performance. A machine learning algorithm trained in a subset of five correct and five incorrect trials using the 65-Hz frequency range window predicted the outcome of remaining trials on a trial-by-trial basis with an accuracy significantly greater than chance (*Figure 2I*, 68% accuracy, *n* = 54 accurately predicted outcomes of 78 tested trials; binomial test, p = 0.0005).

This task-induced increase in 65-Hz frequency range power in the choice period of correct, but not incorrect, trials relative to baseline was also measured in animals that experienced a developmental inhibition of PFC PV interneurons (*Figure 2G, H*, 69 baseline/41 correct/28 incorrect trials from 4 Dev Inhibition animals, mixed effects linear regression, correct versus baseline p = 0.0028, incorrect versus baseline, p = 0.421). However, in contrast to control mice, 65-Hz frequency range power did not significantly differ between correct and incorrect trials in Dev Inhibition mice (*Figure 2H*, mixed effects linear regression, correct versus incorrect, p = 0.0902). Finally, 65-Hz frequency range gamma power during the choice period was not able to predict trial outcome with an accuracy significantly greater than chance in Dev Inhibition mice (*Figure 2I*, 31% accuracy (20/64 trials), binomial test, p = 0.999).

These effects may also involve slightly higher frequencies (65–80 Hz; see *Figure 2—figure supplement 3*) but are unique to what is generally considered the gamma frequency range. Although the 12- to 40-Hz frequency range also predicted task outcome in controls (*Figure 2—figure supplement 3*, 72% (56/78), p = 0.0001), this was also observed in developmental inhibition mice (*Figure 2—figure supplement 3*, 66% accuracy (42/64 trials), p = 0.008). This suggests that specifically the deficit in task-induced 65-Hz frequency range power may contribute to the impairments in ED set-shifting observed in these mice. No other frequency window predicted task outcome in either controls (*Figure 2—figure supplement 3* and 1–4 Hz, 56% (44/78), p = 0.154; 4–12 Hz, 51% (40/78), p = 0.455; 90–120 Hz, 28% (22/78), p = 0.999; 120–200 Hz, 57% (45/78), p = 0.106) or Dev Inhibition mice (*Figure 2—figure supplement 3* and 1–4 Hz, 41% (26/64), p = 0.948; 4–12 Hz, 55% (35/64), p = 0.266; 90–120 Hz, 42% (27/64), p = 0.916; 120–200 Hz, 36% (23/64), p = 0.992).

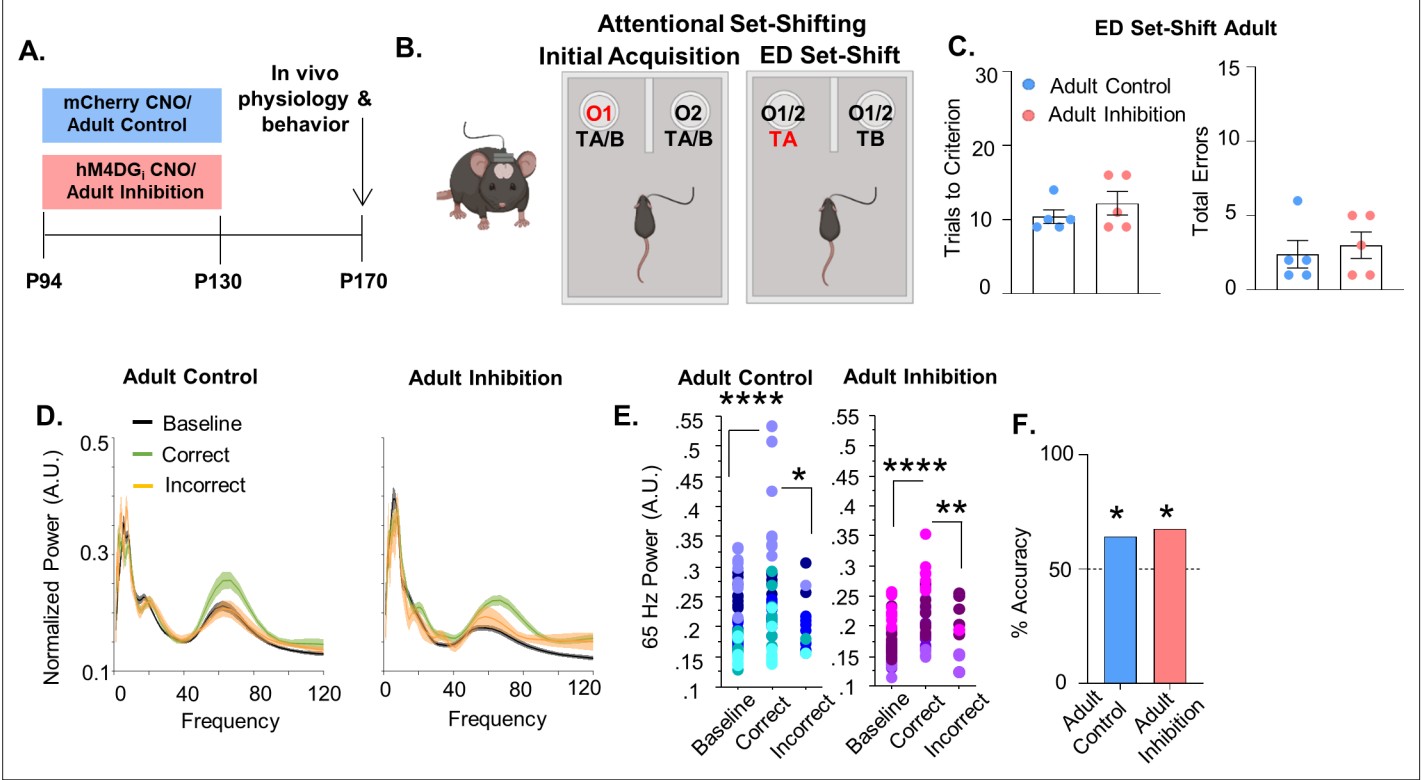

**Figure 3.** Adult inhibition of medial PFC (mPFC) parvalbumin (PV) interneurons does not result in persistent alterations in prefrontal network functioning and behavior in adulthood. (**A**) Clozapine-*n*-oxide (CNO) was administered between P94 and P130 to mice-expressing hM4DG$_i$-mCherry or mCherry in mPFC PV interneurons to produce Adult Inhibition or Control mice, respectively. At P170, mice were evaluated in an attentional set-shifting task while local field potentials were simultaneously recorded in the mPFC as illustrated in the schematic in (**B**). (**C**) Adult Inhibition (*n* = 5) and Control (*n* = 5) mice take a comparable number of trials to reach criterion (left) and make a comparable number of errors (right) during extradimensional (ED) set-shifting. Dots indicate individual animal responses and bars indicate mean ± standard error of the mean (SEM). (**D**) Normalized power (artificial units, A.U.) as a function of frequency in the 3 s prior to when the Adult Control (left, *n* = 51 baseline/39 correct/12 incorrect trials from 5 mice) or Adult Inhibition mice (right, *n* = 44 baseline/34 correct/10 incorrect trials from 4 animals) make a correct (green) or an incorrect (orange) choice relative to baseline (black). Lines and shading indicate mean ± SEM. (**E**) 65-Hz frequency range power is significantly increased prior to correct choices relative to both the baseline period and the choice period prior to incorrect choices in Adult Control (left) and Adult Inhibition mice (right). (**F**) Choice period 65-Hz frequency range power can predict trial outcome in both Adult Control (*n* = 27 out of 42 accurately predicted trial outcomes from 5 mice) and Inhibition animals (*n* = 23 out of 34 accurately predicted outcomes from 4 mice) with an accuracy significantly greater than chance. Bars indicate accuracy of the models. Dotted line indicates chance level (50%). Significance evaluated by unpaired *t*-test (**C**), mixed-effects linear regression (**E**) and binomial test (**F**). *p < 0.05, **p < 0.01, ****p < 0.0001.

The online version of this article includes the following source data and figure supplement(s) for figure 3:

**Source data 1.** Behavioral data relevant to *Figure 3*.

**Source data 2.** 65-Hz frequency range power for adult control mice relevant to *Figure 3*.

**Source data 3.** 65-Hz frequency range power for adult inhibition mice relevant to *Figure 3*.

**Source data 4.** Power predictor accuracy values relevant to *Figure 3*.

**Figure supplement 1.** Additional behavioral analysis related to medial PFC (mPFC) parvalbumin (PV) adult inhibition.

**Figure supplement 1—source data 1.** Behavioral data relevant to *Figure 3—figure supplement 1*.

## Adult inhibition of mPFC PV interneurons does not lead to persistent alterations in behavior and prefrontal network function

To determine whether P14–50 is a sensitive time window we inhibited mPFC PV interneurons for a comparable length of time in mature mice (P94–130, referred to as 'Adult Inhibition and 'Adult Control', respectively; *Figure 3A*). We choose P94–130 because after P90 traditional markers of PFC maturation have stabilized, including the maturation of PV interneurons (*Chini and Hanganu-Opatz, 2020*, *Delevich et al., 2018*; *Miyamae et al., 2017*; *Yang et al., 2014*). Adult Inhibition and Control

mice took a similar number of trials to reach criteria in both the IA (*Figure 3—figure supplement 1*, Adult Control: mean ± SEM: 11 ± 1.18 trials, *n* = 5 mice; Adult Inhibition: 10.4 ± 0.68 trials, *n* = 5 mice; unpaired *t*-test, p = 0.6716) and ED set-shifting portion of the task (*Figure 3C* left; Adult Control: mean ± SEM: 10.4 ± 0.93 trials, *n* = 5 mice: Adult Inhibition: 12.2 ± 1.59 trials, *n* = 5 mice; unpaired *t*-test, p = 0.3576). They also performed a comparable number of errors during ED set-shifting (*Figure 3C*, right, Adult Control: mean ± SEM: 2.4 ± 0.93 errors, *n* = 5 mice; Adult Inhibition: 3 ± 0.89 errors, *n* = 5 mice; unpaired *t*-test, p = 0.6539). There were no differences in the distribution of error types (perseverative or random, *Figure 3—figure supplement 1B, C*; Adult Control: mean ± SEM: 90 ± 10% perseverative, *n* = 5 mice; Adult Inhibition: 66.67 ± 14.3% perseverative, *n* = 5 mice; unpaired *t*-test, p = 0.2179; Adult Control: 10 ± 10% random, *n* = 5 mice; Adult Inhibition: 33.4 ± 14.3% random, *n* = 5 mice; unpaired *t*-test, p = 0.2175) and the latency to complete IA trials (*Figure 3—figure supplement 1D*; Adult Control: mean ± SEM: 106 ± 29.56 s, *n* = 5 mice; Adult Inhibition: 91.64 ± 17.65 s, *n* = 5 mice; unpaired *t*-test, p = 0.688) and ED set-shifting trials (*Figure 3—figure supplement 1E*; Adult Control: mean ± SEM: 47.54 ± 10.29 s, *n* = 5 mice; Adult Inhibition: 109.2 ± 53.32 s, *n* = 5 mice; unpaired *t*-test, p = 0.2887) was equivalent between the groups.

To ensure that these findings were not a consequence of decreased viral expression over time, we injected a second cohort of mice with the virus in adulthood and again saw no impact of adult inhibition on ED set-shifting behavior; trials to criterion were equivalent between Adult Inhibition and Control mice (*Figure 3—figure supplement 1H*; Adult Control: mean ± SEM: 10 ± 0.52 trials, *n* = 12 mice; Adult Inhibition: 10.18 ± 0.66 trials, *n* = 11 mice; unpaired *t*-test, p = 0.8293) as were total errors (*Figure 3—figure supplement 1I*, Adult Control: mean ± SEM: 1.75 ± 0.37 errors, *n* = 12 mice; Adult Inhibition: 1.81 ± 0.50 errors, *n* = 11 mice; unpaired *t*-test, p = 0.9131).

In both Adult Control and Inhibition mice, 65-Hz frequency range power during the choice period significantly differed between correct and incorrect trials (*Figure 3D, E*; 51 baseline/39 correct/12 incorrect trials from 5 Adult Control mice, mixed effects linear regression, correct versus incorrect p = 0.0118; 44 baseline/34 correct/10 incorrect trials from 4 Adult Inhibition mice, mixed effects linear regression, correct versus incorrect, p = 0.0015). Moreover, in both Adult Control and Inhibition groups, 65-Hz frequency range power predicted trial outcome on a trial-by-trial basis with an accuracy significantly greater than chance (*Figure 3F*, Adult Control: 64% (27/42); binomial test, p = 0.0442; Adult Inhibition: 68% (23/34); p = 0.029).

## Developmental inhibition of mPFC PV interneurons results in persistent reductions in their functional inhibition of glutamatergic pyramidal cells in adulthood

Both ED set-shifting behavior and gamma power depend on PV interneuron function. To determine whether developmental inhibition of PV interneurons impairs their functional integration into prefrontal circuitry, mice-expressing channelrhodopsin2 (ChR2) and hM4DG$_i$ in PV interneurons and treated with either CNO ('Dev Inhibition') or Saline ('Dev Control') from P14 to P50, were used for slice electrophysiology at P90 (*Figure 4A, B*). The strength of GABAergic transmission from PV interneurons onto layer II/III pyramidal cells was measured by evoking neurotransmitter release from mPFC PV cells using blue light and recording the amplitude of the resulting light-evoked inhibitory post-synaptic currents (Le-IPSCs) in patch-clamped pyramidal cells. Le-IPSC amplitudes were significantly reduced after the developmental inhibition (*Figure 4C, D*, Dev Control: mean ± SEM: 2023 ± 367 pA, *n* = 11 cells (3 mice); Dev Inhibition: 1141 ± 205.3 pA, *n* = 17 cells (4 mice); unpaired *t*-test, p = 0.032). This decrease in PV interneuron-mediated inhibition was also reflected as a significant reduction the frequency of spontaneously occurring inhibitory post-synaptic currents (sIPSCs) after developmental inhibition (*Figure 4E, F*, Dev Control: mean ± SEM: 4.81 ± 0.47 Hz, *n* = 12 cells (3 mice); Dev Inhibition: 3.37 ± 0.34 Hz, *n* = 17 cells (4 mice); unpaired *t*-test, p = 0.0165). In contrast, sIPSC amplitude was unchanged (*Figure 4G*, Dev Control: mean ± SEM: 49.76 ± 5.69 pA, *n* = 12 cells (3 mice); Dev Inhibition: 50.37 ± 2.66 pA, *n* = 17 cells (4 mice); unpaired *t*-test, p = 0.8885). Notably, adult inhibition of prefrontal PV interneurons did not persistently alter either the frequency (*Figure 4—figure supplement 1C*, Adult Control: mean ± SEM: 2.821 ± 0.954 Hz, *n* = 9 cells (2 mice); Adult Inhibition: 1.857 ± 0.458 Hz, *n* = 10 cells (2 mice); unpaired *t*-test, p = 0.36) or the amplitude (*Figure 4—figure supplement 1D*, Adult Control: mean ± SEM: 37.87 ± 3.63 pA, *n* = 9 cells (2 mice); Adult Inhibition: 44.83 ± 4.77 pA, *n* = 10 cells (2 mice), unpaired *t*-test, p = 0.27) of sIPSCs recorded at P170.

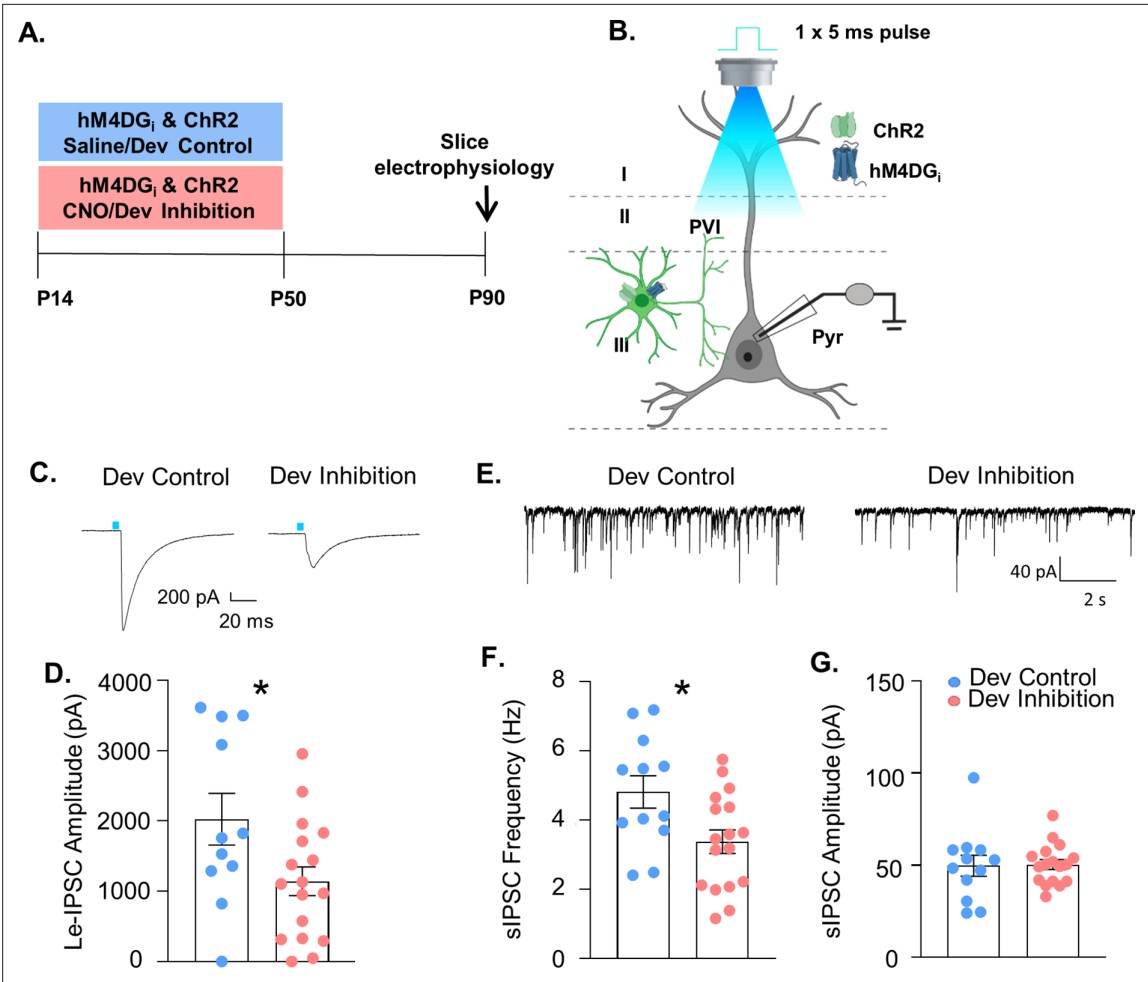

**Figure 4.** Developmental inhibition of medial PFC (mPFC) parvalbumin (PV) interneurons results in persistent reductions in their functional inhibition of glutamatergic pyramidal cells in adulthood. (**A**) Experimental timeline. Mice-expressing channelrhodopsin2 (ChR2) and hM4DG$_i$ in PV cells were administered either clozapine-*n*-oxide (CNO, Dev Inhibition) or Saline (Dev Control) from P14 to P50. At P90 mice were used for slice electrophysiology. (**B**) Experimental schematic. Whole-cell patch clamp recordings were made from pyramidal cells in layer II/III of mPFC from Dev Inhibition and Control mice. The strength of GABAergic transmission from PV cells onto pyramidal cells was measured by evoking neurotransmitter release from mPFC PV cells by delivering a 5-ms pulse of 473 nm blue light via the ×40 objective and recording the amplitude of the resulting light-evoked inhibitory post-synaptic currents (Le-IPSCs). Spontaneous inhibitory post-synaptic currents (sIPSCs) were also recorded. (**C**) Representative traces showing Le-IPSCs from cells recorded from Dev Inhibition and Control mice. (**D**) Le-IPSC amplitudes were significantly reduced in Dev Inhibition mice (*n* = 17 cells from 4 mice) relative to Dev controls (*n* = 11 cells from 3 mice). Dots indicate individual cell responses and bars indicate mean ± standard error of the mean (SEM). (**E**) Representative traces showing sIPSCs from cells recorded from Dev Inhibition and Control mice. (**F**) sIPSC frequency is significantly reduced in Dev Inhibition mice (*n* = 17 cells from 4 mice) relative to Control mice (*n* = 12 cells from 3 mice). Dots indicate individual cell responses and bars indicate mean ± SEM. (**G**) sIPSC amplitude is unchanged (*n*'s indicated in F). Dots indicate individual cell responses and bars indicate mean ± SEM. Significance evaluated with unpaired *t*-tests. *p < 0.05.

The online version of this article includes the following source data and figure supplement(s) for figure 4:

**Source data 1.** Slice electrophysiology data relevant to *Figure 4*.

**Figure supplement 1.** Adult medial PFC (mPFC) parvalbumin (PV) inhibition does not lead to long-lasting effects on their functional inhibition of glutamatergic pyramidal cells.

**Figure supplement 1—source data 1.** Slice electrophysiology data relevant to *Figure 4—figure supplement 1*.

**Figure supplement 2.** Developmental inhibition of medial PFC (mPFC) parvalbumin (PV) interneurons increases GAD65 + puncta but does not alter PV cell number or somatic PV levels.

**Figure supplement 2—source data 1.** Histology data relevant to *Figure 4—figure supplement 2*.

Together, these results indicate that functional inhibition from mPFC PV interneurons onto pyramidal cells is compromised in adulthood following *developmental*, but not *adult*, inhibition of mPFC PV interneuron activity. The change in frequency, but not amplitude, further suggests a decrease in the quantity or functionality of presynaptic inputs.

## Developmental inhibition of mPFC PV interneurons alters the expression of molecular markers

Decreased mPFC PV presynaptic input onto pyramidal cells might result from multiple factors including (1) a loss of PV interneurons, (2) an anatomical alteration in PV interneuron synaptic connections, or (3) a physiological change in PV interneurons that renders them less functional. To address the first possibility, the number of PV-expressing cells was stereologically assessed in the mPFC. The number of PV-expressing cells per mm$^2$ was not altered by the developmental inhibition (*Figure 4—figure supplement 2F, G*; Dev Control: mean ± SEM: 5315 ± 1392 cells/mm$^2$, *n* = 6 mice; Dev Inhibition: 3770 ± 1035 cells/mm$^2$, *n* = 7 mice; unpaired *t*-test, p = 0.3838). To address the second possibility, we stained tissue for GAD65, which is enriched in presynaptic GABAergic terminals, and assessed the number of GAD65 puncta in close proximity to pyramidal cell soma outlined using the cytoskeletal marker, MAP2, as well as the punctal size and fluorescence intensity (*Figure 4—figure supplement 2B*). We found the Dev Inhibition mice had a greater density of GAD65 puncta than Dev Controls (*Figure 4—figure supplement 2C*; Dev Control: mean ± SEM: 0.4382 ± 0.0539 puncta/μm, *n* = 6 mice; Dev Inhibition: 0.5985 ± 0.0301 μm$^2$, *n* = 9 mice; unpaired *t*-test, p = 0.0148), and the puncta were larger (*Figure 4—figure supplement 2D*; Dev Control: mean ± SEM: 0.4074 ± 0.1143 μm$^2$, *n* = 6 mice; Dev Inhibition: 0.8948 ± 0.2368 μm$^2$, *n* = 9 mice; Mann–Whitney, p = 0.0176) and had a greater fluorescence intensity (*Figure 4—figure supplement 2E*; mean ± SEM: Dev Control: 58.86 ± 1.701 artificial units (A.U.), *n* = 6 mice; Dev Inhibition: 64.64 ± 1.432 AU, *n* = 9 mice; unpaired *t*-test, p = 0.0228). To address the third possibility, we histologically assessed somatic levels of PV in PV interneurons. Reductions in levels of PV have been associated with decreased functional inhibition (*Caballero et al., 2020*). We then quantified the intensity of PV staining in virus-infected PV interneurons from Dev Inhibition or Control mice. While the overall mean staining of somatic PV was not changed (*Figure 4—figure supplement 2H*; Dev Control: mean ± SEM: 59.83 ± 4.12 AU, *n* = 113 cells (7 mice); Dev Inhibition: 58.16 ± 1.88 AU, *n* = 377 cells (10 mice); mixed effects linear regression, p = 0.449), there was a non-statistically significant change in the overall distribution of the PV interneuron staining intensities, with an over-representation of lightly stained PV interneurons in the Dev Inhibition group (*Figure 4—figure supplement 2I*; *n* = 113 cells from 7 Dev Control and 377 cells from 10 Dev Inhibition mice, Kolmogorov–Smirnov test, p = 0.0627), which have recently been associated with reduced PV plasticity (*Mukherjee et al., 2019*).

## Enhancing mPFC PV interneuron activity in adulthood with an SSFO rescues the behavioral deficit following developmental inhibition of mPFC PV interneurons

Due to the importance of mPFC PV interneurons for ED attentional set-shifting, we investigated whether we could rescue the deficits in ED set-shifting by acutely and selectively enhancing mPFC PV interneuron excitability in adult mice. To enhance responsiveness of PV interneurons to incoming endogenous activity without imposing an artificial stimulation pattern, we used a stabilized step-function opsin (SSFO) (*Yizhar et al., 2011*). Dev Inhibition or Control mice also expressing the SSFO in their mPFC PV cells were evaluated in the ED set-shifting task in adulthood. On the first day of testing, half the animals had the SSFO activated by administration of 473-nm light delivered via bilaterally implanted optic fibers during ED set-shifting, while the other half were in the light OFF condition. Ten days later, ED set-shifting testing was repeated with the SSFO being activated in the animals that had previously been in the light OFF condition, and vice versa (*Figure 5A–C*). Light activation of mPFC PV interneurons led to a significant reduction in trials to criterion only in Dev Inhibition, but not Dev Control, mice (*Figure 5D*; Dev Inhibition: mean ± SEM: 15.5 ± 1.52 trials light OFF and 10.5 ± 0.85 light ON, *n* = 6 mice; Dev Control: 10.2 ± 0.74 trials light OFF and 10 ± 0.84 light ON, *n* = 5 mice; two-way rmANOVA, fixed effect of light $F_{(1,9)}$ = 14.18, p = 0.004; no fixed effect of treatment $F_{(1,9)}$ = 4.468, p = 0.0637; light by treatment interaction $F_{(1,9)}$ = 12.08, p = 0.007; Bonferroni post hoc Dev Inhibition light ON versus OFF, p = 0.0009).

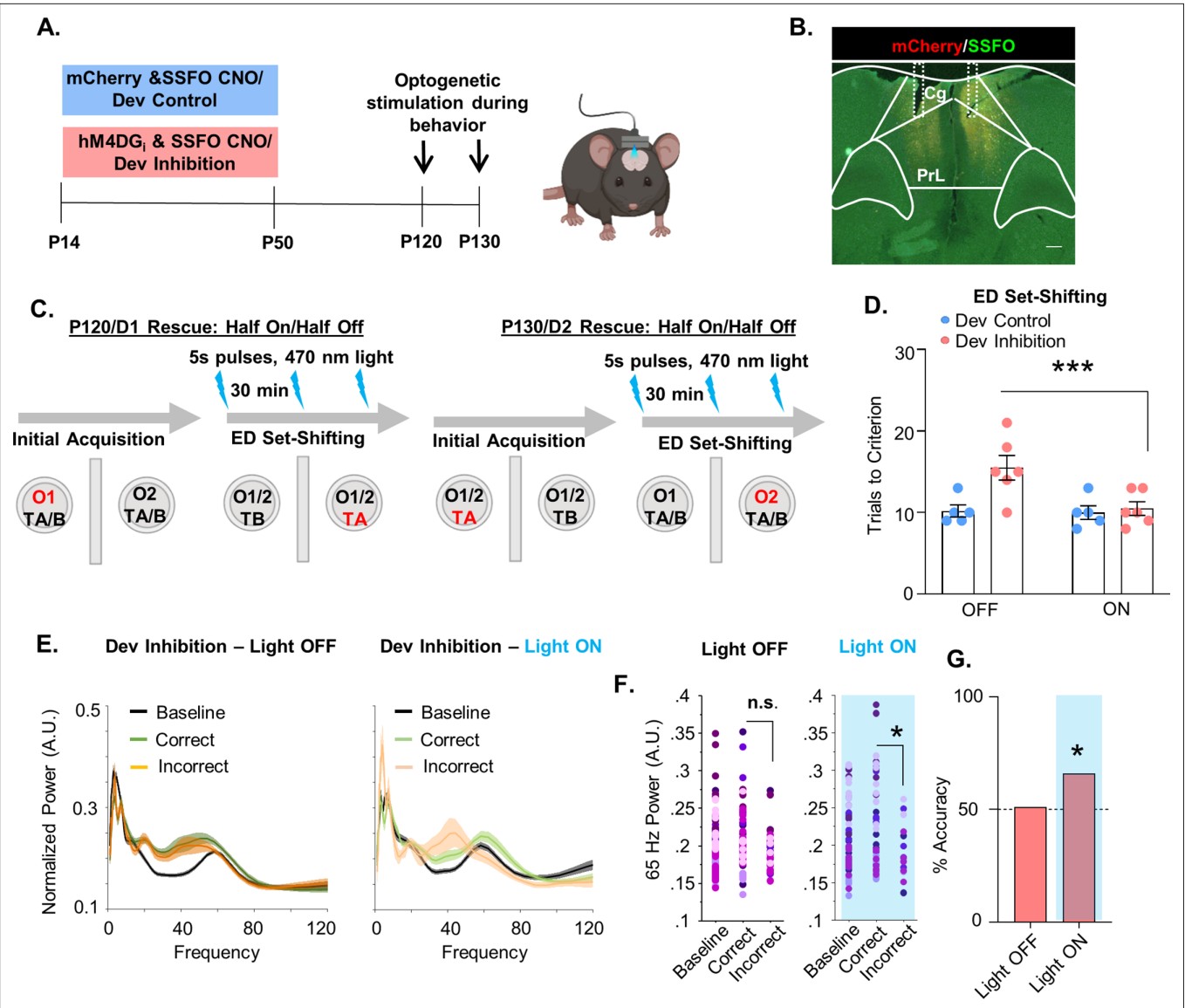

**Figure 5.** Enhancing medial PFC (mPFC) parvalbumin (PV) interneuron activity in adulthood with a stabilized step-function opsin (SSFO) can rescue behavioral deficits following developmental inhibition of mPFC PV interneurons. (**A**) Experimental timeline. Mice expressing an SSFO in combination with hM4DG$_i$-mCherry or mCherry in mPFC PV cells were administered clozapine-n-oxide (CNO) between P14 and P50. In adulthood, optical fibers were bilaterally implanted over the mPFC and mice were evaluated in an attentional set-shifting task with and without optical activation of their PV interneurons. (**B**) Example histology. hM4DG$_i$-mCherry or mCherry (red) and SSFO-EYFP (green) were expressed in PV cells in the mPFC. Dotted lines denote bilateral placement of the fiberoptic implants. (**C**) Schematic illustrating the details of the cross-over experiment. Half the animals had the SSFO activated by administration of 473 nm light via bilaterally implanted optical fibers during extradimensional (ED) set-shifting on testing day 1 with the other half of the animals in the light off condition. Ten days later, testing was repeated with those animals that originally were in the light off condition receiving SSFO activation and vice versa. (**D**) Following light activation of mPFC PV cells, there was a significant reduction in the number of trials it took the Dev Inhibition animals (n = 6) to reach criterion in ED set-shifting but no change in Dev Controls (n = 5). (**E**) Power versus frequency for the choice period prior to correct (green) or incorrect (orange) choices, or in the baseline (black) period for Dev Inhibition animals when the light was OFF (left, n = 92 baseline/61 correct/31 incorrect trials from 6 mice) or ON (right, n = 63 baseline/49 correct/14 incorrect trials from 6 mice). (**F**) Choice period 65-Hz frequency range power is not statistically larger in correct versus incorrect trials in Dev Inhibition animals when the light is OFF (left) but when the light is ON choice period 65-Hz frequency range power is statistically greater in correct versus incorrect trials. (**G**) Choice period 65-Hz frequency range power can predict trial outcome in Dev Inhibition animals with an accuracy significantly greater than chance only when the light is ON (n = 35 out of 53 accurately predicted outcomes from 6 mice), but not when it is OFF (n = 42 out of 82 accurately predicted outcomes from 6 mice). Significance assessed by two-way analysis of variance (ANOVA) followed by post hoc comparison (**D**), mixed effects linear regression (**F**), and binomial test (**G**). *p < 0.05, ***p < 0.001. Scale bar is 250 μm.

The online version of this article includes the following source data for figure 5:

*Figure 5 continued on next page*

*Figure 5 continued*

**Source data 1.** Behavioral data relevant to *Figure 5*.

**Source data 2.** 65-Hz frequency range power for dev inhibition mice with the light OFF relevant to *Figure 5*.

**Source data 3.** 65-Hz frequency range power for dev inhibition mice with the light ON relevant to *Figure 5*.

We then determined whether light stimulation not only rescues the behavior but also restored the changes in 65-Hz frequency range gamma power and its ability to predict task outcome. We found that light stimulation enhanced the difference in choice period 65-Hz frequency range LFP power between correct and incorrect trials in Dev Inhibition mice. At baseline, when the light was OFF, Dev Inhibition mice did not show a significant difference in 65-Hz frequency range power in the period preceding correct versus incorrect choices (*Figure 5F*; 92 baseline/61 correct/31 incorrect trials from 6 mice; mixed effects linear regression, correct versus incorrect, p = 0.0744). However, during PV neuron excitation, the difference between correct and incorrect trials became statistically significant (*Figure 5F*; 63 baseline/49 correct/14 incorrect trials from 6 mice; mixed effects linear regression, correct versus incorrect, p = 0.0463). Light stimulation also restored the ability of 65-Hz frequency range power to predict trial outcome (*Figure 5G*, Dev Inhibition Rescue light OFF: 51% (42/82); binomial test, p = 0.456; Dev Inhibition Rescue light ON: 66%, (35/53); p = 0.0135). These results show that artificially enhancing mPFC PV interneuron activity is sufficient to rescue ED set-shifting behavior and task-induced 65-Hz frequency range power in Dev Inhibition mice with compromised adult mPFC PV interneuron function.

## Discussion

### Identification of a postnatal sensitive period for mPFC PV interneuron development and behavior

Seminal studies in the visual system first identified sensitive periods in which changes in visual experience result in activity-dependent remodeling of thalamocortical inputs that become hard-wired, leading to long-lasting effects on visual functioning (*Hensch, 2005*, *Antonini and Stryker, 1993*; *Hubel and Wiesel, 1970*). Similar activity-dependent remodeling has been shown for the refinement of inhibitory connections made by PV interneurons within the visual cortex during developmental sensitive periods (*Baho and Di Cristo, 2012*; *Chattopadhyaya et al., 2007*; *Fu et al., 2012*; *Wu et al., 2012*). A major result of our study is that the juvenile/adolescent period is also an activity-dependent sensitive period for prefrontal circuit development. Specifically, we found that reversible inhibition of mPFC PV interneurons has long-lasting effects on their functional connectivity, prefrontal network function, and behavior. The window we have identified for these effects spans the juvenile period into adolescence (P14–50). During this time, multiple aspects of mPFC PV maturation typically occur, including physiological alterations in their intrinsic membrane properties and firing properties, strengthening and refinement of their synaptic contacts within prefrontal circuitry, and increases in expression of activity-dependent proteins like PV itself (*Caballero et al., 2020*; *Goodwill et al., 2018*; *Miyamae et al., 2017*; *Yang et al., 2014*; *Bitzenhofer et al., 2020*). The dynamic nature of changes occurring in mPFC PV cells in this time is consistent with the types of changes occurring in PV cells in visual cortex during its period of peak sensitivity to incoming visual input (*Kobayashi et al., 2015*). In keeping with this concept of sensitive periods, inhibiting PV interneurons for a comparable length of time during adulthood did not result in persistent behavioral abnormalities.

Prior work indicates that stimulating pyramidal neurons during an earlier time window, from P7 to P11, the human equivalent of late in utero/early neonatal development, can lead to alterations in prefrontal circuit function, working memory and social preference that persist into adolescence (*Bitzenhofer et al., 2021*). Here, we show that even though the juvenile–adolescent PFC can support complex cognitive tasks, *endogenous* PV interneuron activity during this time remains crucial for its continuing development. Moreover, the cellular and synaptic changes underlying the effects of altering activity in these two different windows differ, as the P7–11 inhibition resulted in changes in PV cell number, which were not affected after P14–50 inhibition.

At the behavioral level we found that reversibly decreasing the activity of mPFC PV interneurons during P14–50 persistently impaired adult PV function and ED attentional set-shifting, consistent with

the known dependence of ED set-shifting on mPFC PV activity (*Canetta et al., 2016*; *Cho et al., 2015*; *Goodwill et al., 2018*; *Cho et al., 2020*). In related findings, Mukherjee et al. found that *increasing* activity of mPFC PV interneurons during a late adolescent window (P60–70) led to persistent rescue of set-shifting behavior in the 22q11 genetic developmental risk factor mouse model (*Mukherjee et al., 2019*). However, that study left unclear if this intervention simply rectified a genetically induced deficit or reflected a general developmental sensitive period. If inhibiting endogenous PV activity during their late adolescent window also elicited long-lasting changes in set-shifting behavior, it would be important to understand if the cellular and synaptic changes underlying that behavioral effect are comparable or disparate to those we see following PV inhibition during the juvenile and early adolescent window. Mukherjee et al. found that a decrease in PV expression was associated with a decrease in excitatory inputs to PV cells (*Mukherjee et al., 2019*). We only observed a non-statistically significant decrease in PV adult expression. Whether this contributes to the decreased inhibition provided by PV neurons to neighboring pyramidal cells remains to be shown. Nevertheless, it is tempting to speculate that while the function of PV interneurons may be compromised following alterations in either window, the specific synaptic processes that are disrupted may differ.

The initial window identified in our experiments is longer than a typical sensitive period in sensory cortex. We do not know whether the CNO-mediated inhibition of PV activity is the same across the entire experimental window. In the developing mediodorsal thalamus, chronic CNO administration during a comparable developmental time window does not impact the ability of CNO to acutely hyperpolarize hM4DG$_i$-expressing neurons (*Benoit et al., 2022*). It is possible hM4DG$_i$ may desensitize in response to chronic agonist activation but such desensitization is believed more likely when expression levels and hM4DG$_i$ receptor reserves are low (*Carvalho Poyraz et al., 2016*; *Roth, 2016*), while the virally mediated expression employed here is high. Future studies will allow for narrowing down the time window during which PV interneuron inhibition produces long-term cognitive deficits.

## Identification of a task-induced gamma signal predictive of ED set-shifting performance

How prefrontal PV interneurons influence prefrontal network function to enable ED set-shifting behavior remains an open question. We found that in control mice, the power of 65-Hz frequency range gamma activity in the mPFC increases while animals are preparing to make a choice in an ED set-shifting task, relative to a baseline period when they first enter the task. Importantly, this increase is only seen in trials where they go on to make a correct, rather than an incorrect, choice. Prefrontal gamma frequency power is believed to reflect organization of the local pyramidal cell activity by PV interneurons. Therefore, the observed task-induced increase in prefrontal gamma power suggests that mPFC PV interneurons are normally engaged while the animal is preparing to make its choice in the ED set-shifting task. Interestingly, recent work from Cho et al. using in vivo calcium imaging to observe bulk changes in mPFC PV interneuron activity during a comparable ED set-shifting task, suggests that mPFC PV cells are activated in incorrect trials, once the animals *realize* they have made an incorrect choice (*Cho et al., 2020*). We examined this in our dataset by looking at whether 65-Hz gamma frequency power increased in the period after, relative to the period before, making either correct or incorrect choices (*Figure 2—figure supplement 4*). In control animals, we found that 65-Hz frequency range power was significantly greater before versus after making a correct choice (*Figure 2—figure supplement 4A, B*). For incorrect trials, 65-Hz frequency range power was larger after versus before making the choice, although the effect did not reach statistical significance (*Figure 2—figure supplement 4A, B*). In Dev Inhibition mice, gamma power did not change before versus after making either a correct or incorrect choice (*Figure 2—figure supplement 4C, D*). It may be that mPFC PV interneurons transiently increase in activity following performance of an incorrect, but not correct, choice and that this transitory peak is better captured with cell-type selective fiber photometry, while gamma oscillations reflect changes in both PV interneurons and other elements of the local prefrontal network. Together, our results and those of Cho et al. provide two potential ways by which mPFC PV interneurons regulate set-shifting behavior.

65-Hz frequency range power increased before correct choices in all control and experimental cohorts. However, while in control groups the magnitude of 65-Hz frequency range gamma power significantly differed between correct and incorrect trials during the choice period, this difference was not observed in Dev Inhibition mice. This decrease in the contrast between correct and incorrect

trials may explain why 65-Hz frequency range power predicts trial outcome in the three control groups but not after developmental inhibition of PV interneurons. Notably, stimulation of PV interneurons in adult Dev Inhibition mice rescued the difference in choice period 65-Hz frequency range power and restored the decoding of task outcome and behavioral performance. Cumulatively, our results suggest that mPFC PV interneurons contribute to ED set-shifting behavior by organizing local pyramidal neuron activity in the gamma frequency at the time when the animals prepare to make their choice. This organization allows the flexible grouping/regrouping of neuronal ensembles representing different task contingencies and stimulus environment, which is necessary for adapting to changing stimulus–outcome associations. However, when PV interneurons are inhibited during postnatal development, their functional maturation is impaired affecting their ability to engage the necessary gamma frequency activity when the mice are adults, thus leading to behavioral impairments.

## Postnatal changes in mPFC PV interneuron activity alters their functional integration into cortical circuitry

We also found that developmental suppression of mPFC PV interneuron activity resulted in persistent reductions in the strength of the functional connections made onto local pyramidal cells. This finding is consistent with recent work showing that knockdown of prefrontal PV beginning in early adolescence persistently impaired the cells' inhibitory output in young adulthood (*Caballero et al., 2020*). The loss of inhibitory output following our developmental inhibition was not due to a loss of PV-expressing cells as stereological counts of mPFC PV interneurons were unchanged in Dev Inhibition mice. An analysis of GAD65-expressing perisomatic puncta used as a proxy for putative inhibitory synapses onto pyramidal cells identified an increase in GAD65 + puncta in Dev Inhibition mice. Prior studies in visual cortex found that during development, PV cell activity and subsequent GABA release can regulate both synapse stabilization and elimination depending on the context (*Baho and Di Cristo, 2012*; *Chattopadhyaya et al., 2007*; *Fu et al., 2012*; *Wu et al., 2012*). We speculate that in Dev Inhibition mice, reduced developmental PV cell activity and GABA release results in insufficient synapse elimination and persistence of immature, less-functional synapses. We also identified a non-statistically significant decrease in the somatic expression of PV itself. Future studies will be able to address whether any of the histological alterations observed following PV inhibition during our developmental sensitive window are causally related to the long-term deficits in PV and PFC network function and cognition.

Our findings comport with prior work in visual cortex demonstrating that alterations in the activity of PV interneurons during a developmental sensitive period can impact their functional integration into cortical circuitry (*Chattopadhyaya et al., 2007*; *Fu et al., 2012*; *Baho and Di Cristo, 2012*; *Wu et al., 2012*). In those studies, alterations in PV activity affected both the maturation and stability as well as the elimination of perisomatic inhibitory synapses made by those cells. Our finding that developmental inhibition of prefrontal PV cell activity increases GAD65 + punctal number is consistent with an important role of activity in synapse elimination in this structure as well. Despite this increase in synapse number, physiological measurements indicate PV functional transmission is compromised, suggesting PV activity may also play an important role in proper synaptic maturation in the PFC. Together, our work in PFC, and prior work in visual cortex, indicates that alterations in activity levels of PV cells during discrete developmental sensitive periods, can lead to long-term consequences for the connectivity that these cells make within cortical circuitry. Given that prefrontal network activity is hypothesized to regulate the developmental refinement of prefrontal circuitry and function more broadly, in future studies it will be important to investigate how our inhibition of mPFC PV activity influences the maturation of excitatory cortical connectivity (*Chini and Hanganu-Opatz, 2020*, *Le Magueresse and Monyer, 2013*).

## Activation of mPFC PV interneurons rescues prefrontal network function and ED set-shifting deficits

Acutely enhancing mPFC PV interneuron function in adulthood using an SSFO rescued prefrontal network function and the impaired ED set-shifting behavior induced by developmental inhibition of mPFC PV interneuron activity. Given that our developmental inhibition of mPFC PV interneuron activity could have resulted in changes in other cell populations in different brain regions due to downstream effects of the chronic inhibition, these results underscore the centrality of defects in mPFC PV interneurons for the cognitive sequelae resulting from this developmental inhibition. These

results are consistent with work demonstrating that optogenetically driving mPFC PV interneurons at gamma frequency (40 Hz) rescues ED set-shifting deficits in adult mice carrying a genetic risk factor for interneuron dysfunction (*Cho et al., 2015*). Optogenetic activation of PV interneurons did not improve set-shifting performance or task-evoked gamma in control mice. Our result implies that in control mice, PV neurons are being optimally engaged for task performance and there is very little room for improvement, consistent with the very low numbers of errors these mice make in the task. Future studies can address whether optogenetic PV interneuron improves performance in control mice under more challenging task conditions.

### Relevance to understanding developmental origins of schizophrenia

Alterations in PFC PV interneurons are a hallmark of schizophrenia, a disorder which is also characterized by alterations in cognitive function such as impairments in ED attentional set-shifting (*Hashimoto et al., 2003*; *Lewis, 2014*; *Brown and Tait, 2016*). Although schizophrenia typically emerges in late adolescence or early adulthood, it has a suspected developmental etiology due to a plethora of early environmental risk factors as well as the presence of attenuated symptoms prior to diagnosis (*Lewis and Levitt, 2002*). However, timing and mechanisms underlying this developmental vulnerability to disease remain unknown.

Our results, together with prior work (*Belforte et al., 2010*), indicate that developing prefrontal PV interneurons are particularly susceptible to changes in their activity levels during a juvenile and adolescent window. If a comparable vulnerability exists in humans than these results would suggest that any genetic or environmental risk factor that alters prefrontal PV activity could potentially lead to long-lasting disruptions in the function of these cells as well corresponding impairments in cognitive behaviors like attentional set-shifting. Both behavioral-level manipulations, such as social interactions, and molecular alterations, such as redox dysregulation (*Behrens et al., 2007*; *Steullet et al., 2017*; *Sullivan and O'Donnell, 2012*), can affect PV activity. Consequently, our results indicate that the timing of exposure to risk factors like social isolation and oxidative stress may be important in determining their long-term effects on behavior. Conversely, these findings suggest that therapeutic interventions delivered within this juvenile and adolescent window might prevent later behavioral dysfunction in people at high risk for developing disease. Finally, our rescue experiments using targeted activation of prefrontal PV interneurons give hope that mechanistically oriented interventions can improve developmentally induced behavioral dysfunction in adulthood.

## Materials and methods

### Animals

All animal procedures were approved by Columbia University's Animal Care and Use Committee (Protocol #1520 and 1618). C57BL/6 (Jackson Labs, Stock #000664) mice were mated with Parvalbumin-Cre (PV-Cre, Jackson Stock #008069) mice to produce heterozygous PV-Cre mice used for early viral injection experiments. For optogenetic slice electrophysiology experiments, mice heterozygous for PV-Cre and homozygous for Ai32 (Cre-dependent ChR2 mice, Jackson Labs, Stock #024109) were mated with RC::PDi homozygous mice (Cre-dependent hM4DG$_i$, gift of Susan Dymecki). Animals were fed ad libitum and reared under normal lighting conditions (12/12 light/dark cycle), unless otherwise noted. While we used both male and female mice for behavior, in vivo and in vitro electrophysiology and histology experiments, we did not have sufficient numbers to detect potential sex differences.

### Surgery

Between postnatal day 9 and 12 (P9–12) young mice were anesthetized with a mixture of 40 mg/kg ketamine and 5 mg/kg xylazine and secured in a stereotax outfitted with a mouse pup adaptor using ear cuffs. Viruses, including AAV5-hSyn-DIO-hM4DG$_i$-mCherry (#44362, Addgene, 1.2 × 10$^{13}$ titer), AAV5-hSyn-DIO-mCherry (#50459, Addgene, 4.8 × 10$^{12}$ GC/ml titer) and AAV5-Ef1a-DIO-hChR2(C128S/D156A)-EYFP (4.9 × 10$^{12}$ titer; UNC Vector Core, Chapel Hill, NC, USA), were injected bilaterally, targeting the prelimbic region of the mPFC (AP +0.92, ML +/−0.13, DV −1.45 from skull at Bregma). 0.2 µl of virus was injected at each site over the course of 2 min, followed by a 2-min wait prior to withdrawing the injection pipette. Mice from each litter were assigned to receive either AAV5-hSyn-DIO-hM4DG$_i$-mCherry or AAV5-hSyn-DIO-mCherry with an attempt to balance the number of

animals in each group across litters and sexes. Animals from both treatment groups (injected with AAV5-hSyn-DIO-hM4DG$_i$-mCherry or AAV5-hSyn-DIO-mCherry virus) were housed together to minimize any confounds due to cage allocation.

A similar procedure was used to inject virus in adult mice. Animals were anesthetized with a mixture of 100 mg/kg of ketamine and 5 mg/kg of xylazine. Virus (0.2 µl per site followed by a 2-min wait) was injected in the mPFC with slightly modified coordinates (AP +1.8, ML +/−0.35, DV −2.5 from skull at Bregma). Carprofen (1 mg/ml i.p., Zoetis, Parsippany, NJ, USA) was given for post-surgical analgesia. Implantation of the electrode bundle and/or optical stimulation fibers occurred approximately 1 week prior to behavioral testing (~P80 for Dev inhibition and Control mice and ~P160 for Adult inhibition and Control mice), to ensure sufficient time for recovery. Adult animals were anesthetized with a mixture of 100 mg/kg of ketamine and 5 mg/kg of xylazine (i.p.), and administered 0.05 ml of 1 mg/ml of dexamethasone (subcutaneously; Henry Schein, Melville, NY, USA) prior to surgery, to reduce brain swelling. Bupivacaine (5 mg/ml; Hospira, Lake Forest, IL, USA) was also injected subcutaneously at the injection site as an additional analgesic. The electrode bundle composed of 76-µm tungsten wire for the LFP and 13-µm tungsten wire for the stereotrodes (California Fine Wire, Grover Beach, CA, USA) was implanted unilaterally in the left mPFC (AP +1.8, ML +/−0.35, DV −2.5 from skull at Bregma). The electrode bundle connected to the microdrive (EIB-16 or EIB-32 narrow; Neuralynx, Bozeman, MT, USA) was fixed to a custom 3D-printed stage. Optic fibers were glued to the stage according to bilateral ML coordinates and the electrode bundle was glued to one of the fibers such that it extended 0.02 mm beyond the tip of the fiber.

## Clozapine-*n*-oxide

CNO was obtained from the NIH and stored at −20°C. A 0.1 mg/ml working solution of CNO was prepared in 0.9% sterile saline at room temperature and administered at a concentration of 1 mg/kg (intraperitoneally, i.p.) twice daily from postnatal day 14 (P14) until P50 (Dev Inhibition/Control and Dev Inhibition/Control with SSFO) or P94 until P130 (Adult Inhibition/Control). The motivation for the dosing of the CNO was based on prior work from our group using hM4DG$_i$ to acutely inhibit other brain circuits (*Parnaudeau et al., 2013*; *Parnaudeau et al., 2015*; *Carvalho Poyraz et al., 2016*), where we use 2 mg/kg. As the effects of CNO have been estimated to last about 8 hr (*Krashes et al., 2011*; *Roth, 2016*; *Zhan et al., 2013*) we split this in two injections per day with 1 mg/kg. This protocol has been used by others to chronically inhibit neurons during postnatal development (*Kozorovitskiy et al., 2012*; *Zhan et al., 2013*; *Roth, 2016*). A new stock solution of CNO was prepared every 2 days and stored at room temperature, protected from light.

## Attentional set-shifting

Attentional set-shifting was performed as previously described (*Canetta et al., 2016*). Mice were food deprived until they reached 80–85% of their baseline body weight. They received 1 day of habituation to the set-shifting enclosure (24″ *L* × 11.5″ *W* × 12″ *H*; 10 min foraging for pieces of honey nut cheerios as a reward). That night they were habituated to digging bowls (terracotta cups, 3″ diameter, 0.75″ height) filled with the bedding media (corn cob or torn paper, Vitakraft, Amazon) and the reward (honey nut Cheerios), which was buried in the bedding media. Habituation was performed overnight in the home cage. Mice then received training/shaping days, in which they were presented with two pots containing samples of either the unscented corn cob or torn paper bedding media baited with a buried honey nut Cheerio reward. Each day the animal received five trials, which continued until the animal found the buried Cheerio reward hidden in each pot. When the animal was consistently digging in each pot within 30–60 s following trial initiation, the next day they proceeded to behavioral testing. The first phase of testing was IA in which the mouse was presented with two pots containing a compound stimulus. The compound stimulus contained a combination of one of two bedding media (corn cob or torn paper) combined with one of two odors (paprika or cinnamon) and only one dimension of the stimulus (e.g., odor, specifically cinnamon) predicted the location of the buried Cheerio reward. For the first five trials of IA, the animals were allowed to dig in both pots to aid them in acquiring the rule, but the trials were scored as either correct or incorrect based on the first pot in which they chose to dig. IA ended when the mice performed 8 out of 10 consecutive trials correctly. The total number of trials to reach this criterion, the number of errors as well as the average latency per trial was analyzed. Once the mouse reached criteria in the IA portion of the task, it proceeded

to the ED set-shifting portion of the task. This phase of the task was identical to IA, except that the dimension of the stimulus that predicted the location of the food reward was changed (e.g., from odor to bedding medium). When the animal performed 8 of 10 consecutive trials correctly, testing was finished. For all trials, if the animal took longer than 10 min to make a choice in a given trial, the trial was ended and recorded as an error. During ED set-shifting, errors resulting from the mouse making the choice that would have been rewarded in IA were called 'perseverative errors'. All other errors were called 'random errors'. For optogenetic stimulation experiments, set-shifting was performed twice. The first session it was performed as above (in IA one odor (cinnamon) was the rewarded stimulus dimension and in set-shifting one bedding medium (torn paper) was the rewarded stimulus dimension). The second session was performed 10 days later but paper bedding was the predictive stimulus in IA and in extra dimensional set-shifting it switched to odor (paprika). Mice performed one session with light ON and one with light OFF in a counterbalanced fashion. We did not observe any training effect between sessions 1 and 2 for either hM4DG$_i$ (rm two-way ANOVA, $n$ = 6 mice, main effect of light $F(1,11)$ = 6.786, p = 0.0245; no main effect of session $F(1,11)$ = 0.1616, p = 0.6954) or mCherry mice (rm two-way ANOVA, $n$ = 5 mice, no main effect of light $F(1,9)$ = 0.0438, p = 0.8384; no main effect of session $F(1,9)$ = 0.04538, p = 0.8384). The night prior to the second session of set-shifting, mice were re-introduced to pots containing unscented versions of both bedding medias baited with Cheerios in their home cage. No additional habituation or training was performed prior to the second set-shifting test. During testing, the investigator was blind to the group status of the animal when possible (e.g., during optogenetic stimulation blinding was not possible).

## Open field

Mice were placed in the center of an automated version of Open Field (OF, KinderScientific) and their movements were measured using infra-red beam technology for 60 min.

## In vivo electrophysiology

In vivo electrophysiology recordings were performed while the animals were performing the set-shifting task. Field potential signals from the mPFC were referenced against a screw implanted in the anterior portion of the skull above the olfactory bulb. LFPs were amplified, bandpass filtered (1–1000 Hz) and acquired at 2000 Hz with Lynx 8 programmable amplifiers on a personal computer running Cheetah data acquisition software (Neuralynx). The animal's position was obtained by overhead video tracking (30 Hz) of a light emitting diode (LED) affixed to the head stage. During testing, TTL signals initiated by the investigator were inserted into the recording when the trial started, when the animal dug, when it ate (if relevant) and when it was removed from the arena were to facilitate later analysis.

Neuralynx files containing LFP data were imported into Matlab with Neuralynx MATLAB import/export package v 4.10. LFP samples were notch filtered using the MATLAB Chronux package to remove 60 cycle noise (http://chronux.org/; rmlinesmovingwinc.m). Mechanical artifacts were eliminated by removing samples whose voltage were more than 3 standard deviations from the entire signal mean. The cleaned signal was then root-mean-squared. Heat maps of normalized power (in A.U.) as a function of frequency and time relative to the dig were constructed using the wavelet transformation package in Matlab (https://www.mathworks.com/help/wavelet/ref/cwt.html). Normalized power as a function of frequency was plotted by averaging the data from these plots in the relevant time windows (e.g., 3 s before dig for the choice period and the first 3 s of the trial for baseline). Any trials less than 6 s long were excluded from the analysis because in these trials the choice and baseline periods would overlap. This resulted in exclusion of nine trials from Dev Control mice (eight correct, one incorrect), five trials from Dev Inhibition mice (four correct, one incorrect), and one correct trial from Adult Control mice. Also note, only four of the five Dev Inhibition and Adult Inhibition mice used for behavior had usable data for physiology. 65-Hz frequency range power encompassed 62–67 Hz power. Electrode locations were confirmed to be within the mPFC based on location of electrolytic lesions.

## Optogenetic stimulation

Optical stimulation was provided by a laser emitting blue light (473 nm) at 4 mW connected via optical fibers (200 μm, 0.22 NA) to the light fibers (200 μm, 0.22 NA, average 80% transmittance) implanted in

the animals' heads to activate the SSFO. Mice were randomized to receive light ON or OFF in a counterbalanced fashion on one of the two set-shifting test days. During testing, mice received a 5-s pulse of blue light when they were in the familiar environment just prior to beginning ED set-shifting. Mice then received a 5-s pulse of blue light every 30 min to maintain the SSFO channels in an open state until testing was complete. All light pulses were administered in the familiar environment between trials.

## Slice electrophysiology

Whole-cell current and voltage clamp recordings were performed in layer 2/3 pyramidal cells and fast-spiking PV-expressing interneurons in the prelimbic region of the mPFC. Recordings were obtained with a Multiclamp 700B amplifier (Molecular Devices) and digitized using a Digidata 1440A acquisition system (Molecular Devices) with Clampex 10 (Molecular Devices) and analyzed with pClamp 10 (Molecular Devices). Following decapitation, 300 µM slices containing the mPFC were incubated in artificial cerebral spinal fluid (ACSF) containing (in mM) 126 NaCl, 2.5 KCl, 2.0 $MgCl_2$, 1.25 $NaH_2PO_4$, 2.0 $CaCl_2$, 26.2 $NaHCO_3$, and 10.0 D-glucose, bubbled with oxygen, at 32°C for 30 min before being returned to room temperature for at least 30 min prior to use. During recording, slices were perfused in ACSF (with drugs added as detailed below) at a rate of 5 ml/min. Electrodes were pulled from 1.5 mm borosilicate-glass pipettes on a P-97 puller (Sutter Instruments). Electrode resistance was typically 3–5 MΩ when filled with internal solution consisting of (in mM): 130 K-gluconate, 5 NaCl, 10 4-(2-hydroxyethyl)-1-piperazineethanesulfonic acid (HEPES), 0.5 ethylene glycol tetraacetic acid (EGTA), 2 Mg-ATP, and 0.3 Na-GTP (pH 7.3, 280 mOsm).

### Effects of CNO on resting membrane potential

$hM4DG_i$-mCherry or mCherry-infected PV cells in the mPFC were identified by their fluorescence at ×40 magnification under infrared and diffusion interference contrast microscopy using an inverted Olympus BX51W1 microscope coupled to a Hamamatsu C8484 camera. Resting membrane potential was recorded in current clamp using the K-gluconate intracellular solution detailed above before and after 10 µM CNO was bath applied to the slice.

### Le-IPSC and sIPSC recordings

Pyramidal cells were visually identified based on their shape and prominent apical dendrite at ×40 magnification under infrared and diffusion interference contrast microscopy using an inverted Olympus BX51W1 microscope coupled to a Hamamatsu C8484 camera. Light-evoked postsynaptic inhibitory currents (Le-IPSCs) and spontaneous inhibitory postsynaptic currents (sIPSCs) were recorded in voltage clamp at a holding potential of −70 mV using a high-chloride intracellular solution containing (in mM): 140 CsCl, 4 NaCl, 1 $MgCl_2$, 10 HEPES, 0.05 EGTA, 2 ATP $Mg^{2+}$, and 0.4 GTP $Mg^{2+}$ (pH 7.3, 280 mOsm). 20 µM 6-cyano-7-nitroquinoxaline-2,3-dione disodium salt (CNQX, Tocris Bioscience, Briston, UK) and 50 µM D-(−)-2-amino-5-phosphonopentanoic acid (AP5, Tocris Bioscience) were added to the bath to block glutamatergic currents. The cells were placed in the center of the field of view, held at −70 mV in voltage clamp and the current response evoked by a 5-ms pulse of blue light (473 nm) applied by an LED (Cool LED, Andover, UK) was recorded. The intensity of the LED was set at 1% of maximum intensity. The current trace was filtered with an eight-pole low-pass Bessel filter and the difference between the baseline and the maximum light-evoked current response was recorded. sIPSCs were assessed from 60 s of the current recording at a holding potential of −70 mV filtered with an eight-pole low-pass Bessel filter and detected using MiniAnalysis (Synaptosoft, Fort Lee, NJ, USA). All event data were averaged by cell.

### Histology

Adult mice were deeply anesthetized with 100 mg/kg ketamine and 5 mg/kg xylazine (i.p.). For in vivo electrophysiology experiments, electrolytic lesions were induced at each recording site by passing current (50 µA, 30 s) through electrodes prior to perfusion. All animals were perfused with phosphate-buffered saline (PBS) followed by 4% paraformaldehyde in PBS. Brains were dissected out and post-fixed in 4% paraformaldehyde overnight before being transferred to PBS for long-term storage. Brains were sectioned serially at 50 µm on a vibratome (Leica, Buffalo Grove, IL, USA). For immunostaining, floating sections were first incubated in PBS containing 2% Triton X-100 for 30 min. They were then

blocked for 1 hr at room temperature with PBS containing 2% Triton X-100 plus 5% fetal calf serum and bovine serum albumin. They were then incubated with primary antibody diluted in blocking buffer for 48 hr at 4°C. The following primary antibodies were used: PV (Sigma, Saint Louis, MO, USA, P3088, 1:2000), glutamate decarboxylase 65 (GAD65; Millipore, Billerica, MA, USA, MAB351, 1:1000), microtubule-associated protein (MAP2; Abcam, ab5392, 1:5000), mCherry (rabbit-anti-dsRed; Takara Bio, Mountainview, CA, USA; 632496, 1:500), or green fluorescent protein (GFP; Abcam, Cambridge, UK, ab13970, 1:1000). After three 15 min washes with PBS containing 2% Triton X-100, the sections were incubated with the relevant Alexa Fluor-conjugated secondary antibodies (Invitrogen, 1:1000) diluted in blocking buffer for 1 hr at room temperature. Sections were rinsed at least three times in PBS, followed by 50 mM Tris pH 7.4 prior to being mounted on SuperfrostPlus slides, coverslipped with mounting media containing Vectashield (info) and stored at 4°C. Stereology was used to assess PV cell number, as well as PV and virus coexpression, in the prelimbic region of adult Dev Inhibition and Control offspring using StereoInvestigator software (MBF Biosciences, Williston, VT, USA) on a Zeiss epifluorescence microscope (Carl Zeiss Microscopy, LLC, White Plains, NY, USA). Levels of PV present in hM4DG$_i$-mCherry and mCherry-expressing cells in the prelimbic region of the mPFC were estimated from analysis of images acquired with a ×10 objective (NA 0.4) on a Leica confocal microscope (Leica Microsystems, Buffalo Grove, IL, USA) from sections stained for PV and mCherry. The viral marker, mCherry was used to identify the confocal plane, and single confocal-plane images were subsequently acquired for both PV and mCherry with PV acquisition settings held constant for all the images. For PV intensity analysis, virus-infected cells were outlined using the polygon tool in ImageJ using the mCherry staining. This outline was transferred to the PV image and the average intensity within that outline was measured for each cell using the Analyze Measurements tool in ImageJ. For GAD65 measurements, tissue sections stained for both GAD65 and the cytoskeletal marker, MAP2, were imaged using a ×63 oil immersion objective (NA 1.4) on a Leica confocal microscope. A single confocal plane was identified using the pyramidal cell marker, MAP2, and then acquired for both MAP2 and GAD65, holding the GAD65 acquisition settings constant for all images. For the GAD65 analysis, a threshold was identified for each image by measuring the background intensity from the interior of 5 pyramidal cell bodies using ImageJ software (NIH) and subtracting this value from 50 (a blinded preliminary assessment of the images identified this metric as a useful one for distinguishing signal from background). A mask was created of all pixels in the GAD65 image above this threshold using the thresholding tool in ImageJ. Pyramidal cells were outlined using the MAP2 stain and this outline was transferred to the GAD65 image, expanded out 2 µm and the 'Analyze Particles' function was used to identify perisomatic puncta (≥2 pixels in size). The number, size, and intensity of each of these puncta were recorded for each cell. Information on 5 cells per image was collected and at least 2 images per animal were used (tissue for some animals was lost to processing). Data were averaged by animal and data for all animals were compared by treatment group. During image acquisition and quantification, the investigator was blind to the treatment.

Viral expression was confirmed from mCherry or GFP staining, and locations of recording site lesions or optical fiber placements were confirmed under DAPI. Mice were excluded from the analysis if they did not have detectable viral expression in the prelimbic region of the PFC or if the stereotrodes and/or optrodes were not localized within this structure.

## Analysis

Statistical analysis and graph preparation of all data except the in vivo electrophysiology were done with Prism 8 software (GraphPad Software, San Diego, CA). In vivo electrophysiology analysis was done with custom scripts in MATLAB (Mathworks, Natick, MA) or R (https://www.r-project.org/) and graph preparation of these data was done in MATLAB or Statview (SAS Institute, Cary, NC, USA). We chose to focus on the 65-Hz frequency range (62–67 Hz) because power in these frequencies maximally distinguished correct from incorrect trials in our Dev Control cohort. Although a number of trials also showed a maximal difference at 58 Hz, we chose to avoid encompassing 60 Hz in our range of assessment. We confirmed that this was the frequency range that was also maximally different between correct and incorrect trials in a second independent control group of mCherry animals given CNO in adulthood (Adult Controls). To analyze differences in 65-Hz frequency range power across animals, trials, and groups, we fit linear mixed models with 65-Hz frequency range power as outcome. The random effect was animal, and fixed effects were trial (baseline, correct, or incorrect) and group

(Dev Control, Dev Inhibition, Adult Control, or Adult Inhibition). For these analyses only data from trials where the latency to make a choice was >6 s were used to avoid an overlap in the time periods considered the choice period and the baseline period. This resulted in the exclusion of nine Dev Control trials (eight correct, one incorrect), five Dev Inhibition trials (all correct), and one Adult Control trial (correct). To analyze 65-Hz frequency range power before versus after making a choice in correct and incorrect trials we looked at the difference in mean 65-Hz frequency range power in the 3 s before and the 3 s after making a choice, split by the trial outcome. We again excluded trials where a choice was made in less than 6 s, resulting in the exclusion of one Dev Control trial (incorrect). To be consistent with the methods of Cho et al., we only analyzed the first five trials of ED set-shifting for this analysis (*Cho et al., 2020*). For predictor analysis, the *fitclinear* command from the MATLAB machine-learning toolbox was used. The 65-Hz frequency range power for each trial was used as a predictor and choice for each trial (correct versus incorrect) was transformed into binary vectors. To avoid over representation of correct trials, equal numbers of either correct or incorrect choice trials were randomly selected to train the model (10 total, 5 correct and 5 incorrect). This model was then tested on the remaining trials and the probability of achieving the observed number of successes given a theoretical distribution based on 50% accuracy was determined with a binomial test. For the predictor analysis, all trials were included regardless of the time it took to make a choice since the choice period was not being compared with the baseline period.

## Sample size determination attentional set-shifting and in vivo electrophysiology

We initially found that Dev Inhibition mice take an average of 20.4 trials to reach criteria while Dev Control animals take 11 trials with an average standard deviation of 4.29. A power analysis suggested a sample size of at least 4 animals per group to be able to reliably detect a significant difference at the level of $p < 0.05$ with a power of 0.8. Therefore, for all subsequent behavioral experiments we included at least 4 animals per group.

## Data sharing plan

All the raw data associated with the figures in this manuscript, as well as R scripts needed to analyze the data, are provided as associated source material.

## Compatibility with reporting standards

Data are reported according to the ARRIVE standards.

# Acknowledgements

The authors thank Julia Greenwald for her assistance with animal husbandry. This work was supported by the Brain and Behavior Research Foundation (grant number 26089) to SEC and (grant number 27384) to AZH and the National Institute of Mental Health (grant number K01MH107760 and R01MH128277) to SEC, (grant number F31 MH119691) to LJB, (grant number K08MH109735) to AZH, and (grant numbers R21 MH121334 and MH117454) to CK. Some figures were created using BioRender.com.

# Additional information

## Funding

| Funder | Grant reference number | Author |
| --- | --- | --- |
| Brain and Behavior Research Foundation | 26089 | Sarah E Canetta |
| Brain and Behavior Research Foundation | 27384 | Alexander Z Harris |
| National Institute of Mental Health | K01MH107760 | Sarah E Canetta |

| Funder | Grant reference number | Author |
|---|---|---|
| National Institute of Mental Health | R01MH128277 | Sarah E Canetta |
| National Institute of Mental Health | F31MH119691 | Laura J Benoit |
| National Institute of Mental Health | K08MH109735 | Alexander Z Harris |
| National Institute of Mental Health | R21MH121334 | Christoph Kellendonk |
| National Institute of Mental Health | R21MH117454 | Christoph Kellendonk |

The funders had no role in study design, data collection, and interpretation, or the decision to submit the work for publication.

### Author contributions

Sarah E Canetta, Conceptualization, Resources, Formal analysis, Supervision, Funding acquisition, Investigation, Methodology, Writing – original draft, Writing – review and editing; Emma S Holt, Laura J Benoit, Formal analysis, Investigation, Writing – review and editing; Eric Teboul, Investigation, Writing – review and editing; Gabriella M Sahyoun, Formal analysis; R Todd Ogden, Formal analysis, Writing – review and editing; Alexander Z Harris, Resources, Formal analysis, Supervision, Funding acquisition, Writing – review and editing; Christoph Kellendonk, Conceptualization, Resources, Supervision, Funding acquisition, Methodology, Writing – original draft, Writing – review and editing

### Author ORCIDs
Sarah E Canetta http://orcid.org/0000-0002-3860-2504
Alexander Z Harris http://orcid.org/0000-0001-9089-0366
Christoph Kellendonk http://orcid.org/0000-0003-3302-2188

### Ethics

All animal procedures were approved by Columbia University's Animal Care and Use Committee (Protocol #1520 and 1618).

### Decision letter and Author response

Decision letter https://doi.org/10.7554/eLife.80324.sa1
Author response https://doi.org/10.7554/eLife.80324.sa2

## Additional files

### Supplementary files

• MDAR checklist

• Source code 1. R script used to analyze 65-Hz freqency range power relevant to *Figures 2 and 3F*. This code is designed provide a linear mixed model analysis of 65-Hz frequency range power, with a fixed effect of trial type (baseline, correct choice, and incorrect choice) and a random effect of animal. It provides comparisons of whether 65-Hz frequency range power varies significantly between trial types (baseline versus correct or incorrect choice as well as correct versus incorrect choice) for the different data sets.

### Data availability

All the raw data associated with the figures in this manuscript, as well as R scripts needed to analyze the data, are provided as source data.

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
