## [Editor Report]

The authors explored the time-dependent effects of inhibition of parvalbumin-expressing interneurons (PV cells) in the mouse prefrontal cortex on task learning and cognition. Overall, the important study provides solid evidence showing that prefrontal cortex PV cell activity during a sensitive period of neurodevelopment affects prefrontal cortex function in the adult mouse. This study progresses our understanding of the development of the prefrontal cortex.

---

## [Decision Letter]

**Decision letter after peer review:**

Thank you for submitting your article "Mature parvalbumin interneuron function in prefrontal cortex requires activity during a postnatal sensitive period" for consideration by *eLife*. Your article has been reviewed by 3 peer reviewers, and the evaluation has been overseen by a Reviewing Editor and Laura Colgin as the Senior Editor. The following individuals involved in the review of your submission have agreed to reveal their identity: Graziella Di Cristo (Reviewer #1); Achira Roy (Reviewer #2).

Essential revisions:

1) Improve detection of PV-expressing synapses, as explained in detail by reviewer 1.

2) New experiments using a more appropriate chemogenetic protocol, also including a locomotion test, checks for CNO-induced depolarization of control cells, as explained by reviewer 3.

3) Discuss the possibility of compensatory changes interacting with critical periods.

4) Revise statistical analysis for various data sets as explained by Reviewer 3.

5) Broaden the data power spectra analyzed.

6) Re-analyze sIPSCs and mIPSCs, as explained by reviewer 3.

7) Implement editorial changes as proposed by all the reviewers.

*Reviewer #1 (Recommendations for the authors):*

This is a timely and interesting study.

My major technical concern about the study relates to the anatomical analysis of differences, or lack thereof, in putative PV cells synapses and PV expression levels. Overall, the manuscript lacks information on how confocal images were acquired (Z-step, frame average, pixel size) and quantified (saying that ImageJ was used is not enough, how was analysis performed exactly? How was the threshold for masking determined?). More importantly, it looks like GAD65 puncta were analyzed in images acquired using a 10X objective. To determine optical resolution the authors need to report the objective NA, nevertheless, a 10X objective usually does not give enough resolution to measure the sizes of objects as small as synapses. It is very likely that potential differences went undetected. Authors also report a number of GAD65 puncta, but these should be normalized on the perimeter of the postsynaptic target. The same concerns apply to how PV cell intensity was determined. Finally, in absence of clearly significant data, the result section subtitle "Developmental inhibition of mPFC PV interneurons results in persistent reductions in the expression of molecular markers" is misleading since no statistically significant results were reported.

Other major issues that require clarifications

– Were control mice injected with CNO daily as well? That would be the correct control. Please, state clearly in the methods how controls were treated.

– Data shown in Figures S2B and S2C are not clear to me. I would expect error types to be increased in Dev inhibition. Please clarify

– Regarding the sentence "Further, the magnitude of the difference in 65 Hz frequency range power during the choice period of correct versus incorrect trials was significantly higher in Dev Control versus Inhibition mice (mixed-effects linear regression, p<0.0001)". I am not sure this statement is supported by Figure 2H.

– Figure S4. Could the authors plot the data for the 65-80Hz range?

– The author state that "Male and female mice were used for behavior, in vivo and in vitro electrophysiology and histology experiments (no effects of sex were seen)". With a N of 4-6, the authors do not have sufficient power to make this statement. Please state that you used both sexes but you have no sufficient N to detect potential differences

– For ex vivo electrophysiology recordings, the author should indicate both N of cells and n of mice. While the field does use multiple cells from a single animal for numerous cellular parameters, this may not be statistically valid. For an example, see Lazic et al., 2010 (PMID: 20074371). Using N=cell leads to an increase in false positives as it overpowers the experiment. The independent replicate in these experiments is the animal (it is a comparison of treatment) and two cells within the same animal are not independent. Ideally, a statistic that takes into account the variability within and between animals is best (linear mixed modelling or generalised linear mixed modelling, depending on whether the data is normally distributed). The same concern applies to data represented in Figure 5H.

– Figure 6F left panel. There is no statistically significant difference between baseline and correct choice. Please compare with Figure 2H and discuss.

*Reviewer #2 (Recommendations for the authors):*

Smoother, logical transitions between paragraphs in the Discussion can make the manuscript easier to read and to understand the analogies and comparisons that the authors made.

The following concerns need to be addressed:

1. Majority of genetic and/or circuit-level perturbations that may lead to neurological disorders such as schizophrenia and autism occur early in embryonic development. Even the parvalbumin-positive interneurons are born and begin to migrate at embryo stages. Authors should clarify the biological significance of choosing the P14 mouse as the earliest observation stage for "developmental inhibition".

2. Figure 5: Authors have used the term "non-statistically significant" for somatic PV levels while the PV cell and synapse number are inferred as non-significantly different. What parameters have the authors assumed while distinguishing these two types of statistically insignificant data? Can the sample size be increased for Figure 5(I) to see if at all this can be statistically significant?

3. Page 13 (Discussion): Last two sentences of the first paragraph are confusing; authors should elaborate on what brain regions with citations. Further, sensitive windows of different processes exist along the entire developmental timeline; so, it is unclear what the authors intended to say here.

*Reviewer #3 (Recommendations for the authors):*

1) The choice of statistical tests and the effect some of the outliers are having on the analysis is questionable. The main concern is that for EEG power, instead of mixed-effects linear regression, one-way ANOVA should be used. In addition, have the authors carried out a test of outliers on their data? For instance, in Figure 3E, are the top two blue data points outliers? If they are, what is the effect of excluding them?

2) The authors have focused on the 65 Hz power but there are changes throughout the spectra. A good example of this is in Figure S4. If inaccuracy (instead of accuracy) is measured, will 90-120 Hz become significant? The authors need to address this issue systematically.

3) From the images provided, the injection does not seem to be contained within the mPFC. Is it possible to quantify the number of neurons that are expressing constructs outside of the mPFC? Moreover, as the authors note that frontal motor areas are affected, is there a difference in locomotion between the CNO-treated control and hM4DGi animals? This is vital as locomotion will impact the behavioral tasks.

4) For figure 1D, what was the rationale for mixing the ages of mCherry controls? This should be two separate bars.

For the same figure, why is CNO depolarizing mCherry controls? Will CNO depolarize principal cells? Most of the extent of depolarization in the Figure will have physiological consequences.

5) Also, from the sample traces of P115 hM4DGi, there is evidence of slow depolarization after peak – over time this may have the potential to become depolarizing.

6) For Figure 1E-F please provide error bars. Importantly, what were the resting membrane potentials of the neurons? This will be important for spike firing.

Why were the spikes not saturated in the step protocol? For 1E the statistical analysis was done at 200 pA and for 1F this was 150 pA? What was the criteria here?

7) Please include the names of ImageJ plug-ins that were used for analysis. If custom java scripts were used, then please state so.

8) sIPSC amplitudes within a cell are rarely normally distributed. Can the authors confirm that their IPSC amplitudes per cell are indeed normally distributed? If not, then please use median or cumulative probability graphs.

Were sIPSCs measured from the same cells as Le-IPSCs? Activating networks using light beforehand will mean that interneuron networks will have been perturbed for spontaneous IPSC measurement.

9) What happens to mIPSCs? Do the authors see a change in mini amplitudes consistent with a reduction of presynaptic quantal release?

10) Since the authors do not see any change in PV neuron / presynaptic terminal staining, the rationale for including the results in Figure 5 is not clear. In addition, in Figure 5H, for the non-parametric KS, control has a third of the cells as dev inhib. The number of cells for these tests should be roughly equal.

For images in Figure 5B/F what were the conditions? Please show sample images for all conditions.

---

## [Author Response]

Essential revisions:1) Improve detection of PV-expressing synapses, as explained in detail by reviewer 1.

We have redone the analysis of the PV-expressing synapses at a higher magnification (63x, NA 1.4) as requested by the reviewer (Figure S7).

2) New experiments using a more appropriate chemogenetic protocol, also including a locomotion test, checks for CNO-induced depolarization of control cells, as explained by reviewer 3.

We now include new open field data demonstrating there are no locomotor differences between the Dev Inhibition and Control animals (new Figure S2F). We also call the reviewers’ attention to the previously included data that the latency to make a choice in the attentional set shifting task is not different between the groups (Figure S2D-E), consistent with the idea that locomotion is not impaired. We also address the reviewer’s concern that CNO depolarizes control cells with a new statistical analysis demonstrating that CNO does not significantly change the resting membrane potential of control cells from 0.

Our approach of twice daily injections of CNO is consistent with that of other publications that have used similar chemogenetic approaches to chronically alter activity^1-3^. However, the reviewer is correct that our twice daily CNO injection protocol may only intermittently inhibit PV cells, and it is possible that persistent inhibition might result in even stronger behavioral and circuit effects. However, it is also possible that more continuous CNO administration could lead to hM4DG_i_ desensitization. Given these caveats, we respectfully submit that repeating all the experiments under conditions that would allow constant chronic dispensation of CNO (such as implantation of minipumps) is an excellent future experiment but currently outside the scope of this manuscript.

3) Discuss the possibility of compensatory changes interacting with critical periods.

Our current studies demonstrate that a 35-day window of inhibition during development, but not during adulthood, leads to long-lasting effects on behavior and prefrontal network function. If the developmental manipulation is more impactful because the manipulation represents a longer proportion of the animals’ lifetime (as suggested by Reviewer 3), we would expect that the effect should wane as the animal gets older. However, for the rescue experiments that take place at P120 and P130, we still find that the Dev Inhibition animals are impaired at baseline, suggesting that it is not the proportion of the animals’ lifetime that has been inhibited, but the timing during which this inhibition occurs, that matters most.

4) Revise statistical analysis for various data sets as explained by Reviewer 3.

Because the data consists of multiple trials collected from each of several different animals, the model should account for the non-independence of observations from the same animal. An appropriate way to model this data is with mixed effects linear regression, using a fixed effect of treatment and a random effect of animal. To examine the effect of potential outliers, we repeated the analysis on log-transformed data which effectively accounts for right-skewness, and the results are similar to the analysis on the original data.

5) Broaden the data power spectra analyzed.

We now plot the data for the 65-80 Hz range in Figure S4 as suggested by Reviewer 1. We find that 65-80 Hz frequency power accurately predicts trial outcome for Dev Control but not Inhibition mice, which is not surprising given that this range overlaps with that used for the 65-Hz frequency range power (62-67 Hz). In control animals, the prediction accuracy is slightly higher for the 65-Hz frequency range (68%) compared with the 65-80 Hz range (60%). We also changed the title of Figure S4 to “Accuracy of trial outcome prediction based on the power of the different frequencies” to clarify for Reviewer 3 that 90-120 Hz power is bad at predicting outcome in line with the power for correct and incorrect choice periods being largely overlapping in that range.

6) Re-analyze sIPSCs and mIPSCs, as explained by reviewer 3.

We analyzed sIPSC frequency and amplitude to assess normality and find that the cell average data is normally distributed based on the Anderson-Darling and Kolmogorov-Smirnov tests, so the use of a t-test to compare the values is appropriate.

7) Implement editorial changes as proposed by all the reviewers.

We have incorporated suggested changes to the text as described in detail below.

Reviewer #1 (Recommendations for the authors):This is a timely and interesting study.My major technical concern about the study relates to the anatomical analysis of differences, or lack thereof, in putative PV cells synapses and PV expression levels. Overall, the manuscript lacks information on how confocal images were acquired (Z-step, frame average, pixel size) and quantified (saying that ImageJ was used is not enough, how was analysis performed exactly? How was the threshold for masking determined?). More importantly, it looks like GAD65 puncta were analyzed in images acquired using a 10X objective. To determine optical resolution the authors need to report the objective NA, nevertheless, a 10X objective usually does not give enough resolution to measure the sizes of objects as small as synapses. It is very likely that potential differences went undetected. Authors also report a number of GAD65 puncta, but these should be normalized on the perimeter of the postsynaptic target. The same concerns apply to how PV cell intensity was determined. Finally, in absence of clearly significant data, the result section subtitle "Developmental inhibition of mPFC PV interneurons results in persistent reductions in the expression of molecular markers" is misleading since no statistically significant results were reported.

Thank you for these suggestions. We have now provided extensive additional details of the image acquisition and analysis parameters as requested in the Methods section on page 26. Based on the reviewer’s suggestion, we repeated the GAD65 perisomatic punctal analysis using images acquired at 63x (NA 1.4). In this new analysis, we found that the total number of GAD65 puncta as well as the intensity and size of the puncta were increased in our Dev Inhibition mice (Figure S7). Although total pyramidal cell perimeter did not differ between treatment groups, we present the data normalized to pyramidal cell perimeter as per the reviewer’s suggestion. Taken together with our electrophysiological results, these histological differences indicate that developmental inhibition of PV interneurons leads to an overabundance of putative perisomatic synaptic contacts that are nonetheless associated with a decrease in overall PV cell functional connectivity. PV interneuron activity and GABA release during development has been associated with both synapse elimination as well as stabilization and maturation depending on the context^4-7^ and so we can speculate that in Dev Inhibition mice, reduced developmental GABA release results in insufficient synapse elimination and persistence of immature, less-functional synapses. Because it remains speculative how this histological phenotype in Dev Inhibition mice relates to impairments in PFC network function and behavior, we have moved this figure to the Supplementary Figures. We added a short paragraph in the discussion regarding this point (p. 1617).

Other major issues that require clarifications– Were control mice injected with CNO daily as well? That would be the correct control. Please, state clearly in the methods how controls were treated.

Yes, control mice were stereotaxically injected with an AAV expressing mCherry and administered CNO on an identical schedule to hM4D-expressing mice. As stated on p.5, ‘we injected P10 PV-Cre pups with an AAV expressing hM4DG_i_ or mCherry and administered CNO between P14 and P50 (subsequently referred to as ‘Dev Inhibition’ or ‘Dev Control’, respectively; Figure 2A)’. This is also depicted in the picture in Figure 2A.

– Data shown in Figures S2B and S2C are not clear to me. I would expect error types to be increased in Dev inhibition. Please clarify

Figure S2B and C present a breakdown of the types of errors made by Control and Dev Inhibition mice, depicted as the percent of errors that are perseverative or random (normalized to the total number of errors). While there is a robust increase in the number of total errors made in Dev Inhibition mice (Figure 2C), the percent of errors which are perseverative and those which are random do not differ between groups. Of note, since there are some mice in the control group who do not make any errors, Figure S2B and C has fewer mice included than Figure 2C.

– Regarding the sentence "Further, the magnitude of the difference in 65 Hz frequency range power during the choice period of correct versus incorrect trials was significantly higher in Dev Control versus Inhibition mice (mixed-effects linear regression, p<0.0001)". I am not sure this statement is supported by Figure 2H.

We tested whether the magnitude of the correct versus incorrect difference statistically differed and reported this difference in the text, but we did not originally depict that difference in the figure because it was not our primary output and we thought it made the figure too complicated. We have removed this statistical comparison from the manuscript to be consistent with what is shown in the figure.

– Figure S4. Could the authors plot the data for the 65-80Hz range?

We now plot the data for the 65-80 Hz range in Figure S4. We find that 65-80 Hz frequency power accurately predicts trial outcome for Dev Control but not Inhibition mice, which is not surprising given that this range overlaps with that used for the 65-Hz frequency range power (62-67 Hz). The prediction accuracy is slightly higher for the 65-Hz frequency range (68%) compared with the 65-80 Hz range (60%).

– The author state that "Male and female mice were used for behavior, in vivo and in vitro electrophysiology and histology experiments (no effects of sex were seen)". With a N of 4-6, the authors do not have sufficient power to make this statement. Please state that you used both sexes but you have no sufficient N to detect potential differences

We have now added that statement (p.19): ‘While we used both male and female mice for behavior, in vivo and in vitro electrophysiology and histology experiments, we did not have sufficient numbers to detect potential differences.’

– For ex vivo electrophysiology recordings, the author should indicate both N of cells and n of mice. While the field does use multiple cells from a single animal for numerous cellular parameters, this may not be statistically valid. For an example, see Lazic et al., 2010 (PMID: 20074371). Using N=cell leads to an increase in false positives as it overpowers the experiment. The independent replicate in these experiments is the animal (it is a comparison of treatment) and two cells within the same animal are not independent. Ideally, a statistic that takes into account the variability within and between animals is best (linear mixed modelling or generalised linear mixed modelling, depending on whether the data is normally distributed). The same concern applies to data represented in Figure 5H.

Whether cells or animals represent the biological unit for independent replicates in slice electrophysiological experiments remains an important question. Analysis of our slice electrophysiological data found that the estimated animal effect was small relative to the noise level, and formal hypothesis testing indicated that a random effect for animal was not required. Therefore, we have left the analysis of the slice electrophysiology data unchanged but now report the number of animals that contributed to each of these measurements in addition to the number of cells. For the analysis of PV cell intensity depicted in the previous Figure 5H (now Figure S7), we did find that a random effect of animal was required. Therefore, the statistical analysis for Figure S7H has been changed to a linear mixed model using treatment as a fixed effect and animal as a random effect.

– Figure 6F left panel. There is no statistically significant difference between baseline and correct choice. Please compare with Figure 2H and discuss.

In all conditions we observe a significant increase in γ frequency power in correct choice versus the baseline period. We did not plot this difference or report it for Figure 6 because our primary focus was on correct choice-period versus incorrect choice-period γ because it is this difference that endows choice period γ power with its capacity to predict trial outcome accurately in controls and that is compromised by developmental inhibition of PV cells. In Dev Inhibition animals also carrying a selective step function opsin in their PV interneurons, there is no significant difference between γ power in the choice period preceding correct and incorrect trials when the light is OFF, but when the light is ON, γ power is significantly higher in correct versus incorrect trials. This is associated with an improved ability of γ power to predict trial outcome.

Reviewer #2 (Recommendations for the authors):Smoother, logical transitions between paragraphs in the Discussion can make the manuscript easier to read and to understand the analogies and comparisons that the authors made.

We edited the discussion and hope that it is now easier to follow.

The following concerns need to be addressed:1. Majority of genetic and/or circuit-level perturbations that may lead to neurological disorders such as schizophrenia and autism occur early in embryonic development. Even the parvalbumin-positive interneurons are born and begin to migrate at embryo stages. Authors should clarify the biological significance of choosing the P14 mouse as the earliest observation stage for "developmental inhibition".

It is a good point that putative PV-expressing neurons go through important developmental stages prior to P14, and we do not wish to imply that developmental manipulations during an earlier window might not result in long-lasting effects. In fact, as we discuss on p.3, prior work manipulating cortical pyramidal cell activity in the neonatal stage resulted in abnormalities in PV interneurons later in life^9^. However, extensive refinement of prefrontal circuitry, and in particular with regard to the development of the functional properties and connections of PV interneurons, continues through adolescence, and whether neuronal activity during this later period is required for prefrontal circuit maturation remained an open question that we sought to address in this study.

2. Figure 5: Authors have used the term "non-statistically significant" for somatic PV levels while the PV cell and synapse number are inferred as non-significantly different. What parameters have the authors assumed while distinguishing these two types of statistically insignificant data? Can the sample size be increased for Figure 5(I) to see if at all this can be statistically significant?

Using a high magnification analysis, as suggested by reviewer 1 we now found that the total number of GAD65 puncta as well as the intensity and size of the puncta were increased in our Dev Inhibition mice.

We have moved the histological data to the supplementary figures and added a short paragraph to the discuss the implications of the new findings (p. 16-17). Moreover, in keeping with the concerns of the reviewers, we no longer emphasize the change in PV intensity as a potential mechanism underlying the functional deficit.

3. Page 13 (Discussion): Last two sentences of the first paragraph are confusing; authors should elaborate on what brain regions with citations. Further, sensitive windows of different processes exist along the entire developmental timeline; so, it is unclear what the authors intended to say here.

We have revised to the following: ‘The dynamic nature of changes occurring to mPFC PV cells in this time is consistent with the types of changes occurring in PV cells in visual cortex during its period of peak sensitivity to incoming visual input^10^. In keeping with this concept of sensitive periods, inhibiting PV interneurons for a comparable length of time during adulthood did not result in persistent behavioral abnormalities.’

Reviewer #3 (Recommendations for the authors):1) The choice of statistical tests and the effect some of the outliers are having on the analysis is questionable. The main concern is that for EEG power, instead of mixed-effects linear regression, one-way ANOVA should be used. In addition, have the authors carried out a test of outliers on their data? For instance, in Figure 3E, are the top two blue data points outliers? If they are, what is the effect of excluding them?

Because the data consists of multiple trials collected from each of several different animals, and data within a given animal are highly correlated with one another, the model should account for the non-independence of observations from the same animal. We believe that the most appropriate way to model this data is with mixed effects linear regression, using a fixed effect of treatment and a random effect of animal. To examine the effect of potential outliers, we repeated the analysis on logtransformed data which effectively accounts for right-skewness, and the results are similar to the analysis on the original data.

2) The authors have focused on the 65 Hz power but there are changes throughout the spectra. A good example of this is in Figure S4. If inaccuracy (instead of accuracy) is measured, will 90-120 Hz become significant? The authors need to address this issue systematically.

We apologize that the title of Figure S4 was confusing. We have changed it to “Accuracy of trial outcome prediction based on the power of different frequencies”. What it shows is that power between 90 and 120 Hz is bad at predicting trial outcome in line with the power for correct and incorrect choice periods being largely overlapping.

3) From the images provided, the injection does not seem to be contained within the mPFC. Is it possible to quantify the number of neurons that are expressing constructs outside of the mPFC? Moreover, as the authors note that frontal motor areas are affected, is there a difference in locomotion between the CNO-treated control and hM4DGi animals? This is vital as locomotion will impact the behavioral tasks.

The reviewer correctly notes that there is some spread outside the mPFC to the adjacent M2 region. We found that on average approximately 70% of the prelimbic region of the mPFC showed viral expression while only 16% of M2 showed expression (new Figure S1C). We now include new data showing that there are no differences between the groups in locomotion in the open field (new Figure S2F), which corroborates data previously depicted that there are no differences in the latency it takes the animals to make a choice in the set-shifting task (Figure S2D-E).

4) For figure 1D, what was the rationale for mixing the ages of mCherry controls? This should be two separate bars.

We have never seen any effect of CNO on resting membrane potential in non-hM4DG_i_-expressing cells (including P35 mCherry-expressing PV cells as well as P35, P50 and P115 mCherry-expressing thalamic cells^2^). These observations were the rationale for combining the mCherry controls recorded at the 2 different ages. However, we have changed the analysis to be a 2-way ANOVA looking at the effects of both age of the tissue (P35 versus P115) and virus used (mCherry versus hM4DG_i_) on CNO-induced change in resting membrane potential. We now show that there is a highly significant main effect of virus (n=5 P35 and 3 P115 mCherry cells and 7 P35 and 6 P115 hM4DG_i_-mCherry cells, 2way ANOVA, F_(1,17)_=9.069, p=0.0079) but no main effect of age (n=5 P35 and 3 P115 mCherry cells and 7 P35 and 6 P115 hM4D cells, 2-way ANOVA, F_(1,17)_=0.1391, p=0.7138) nor any interaction of age and virus (n=5 P35 and 3 P115 mCherry cells and 7 P35 and 6 P115 hM4D cells, 2-way ANOVA, F_(1,17)_=0.9260, p=0.3494). This new analysis is on p.5.

For the same figure, why is CNO depolarizing mCherry controls? Will CNO depolarize principal cells? Most of the extent of depolarization in the Figure will have physiological consequences.

CNO does not statistically change the resting membrane potential from 0 in mCherry-expressing controls (Wilcoxon sign rank test, p>0.9999).

5) Also, from the sample traces of P115 hM4DGi, there is evidence of slow depolarization after peak – over time this may have the potential to become depolarizing.

We believe the reviewer is referring to the slight decay in the peak CNO-induced hyperpolarization that is observed in one of the traces. However, this is not observed consistently (most of the time the hyperpolarization remains stable) and when it does occur, this never becomes a depolarization from the baseline resting membrane potential, even if you extend the window of observation.

6) For Figure 1E-F please provide error bars. Importantly, what were the resting membrane potentials of the neurons? This will be important for spike firing.Why were the spikes not saturated in the step protocol? For 1E the statistical analysis was done at 200 pA and for 1F this was 150 pA? What was the criteria here?

These measurements were made in current clamp starting from the cells’ natural resting potential and the differences in baseline resting potential may therefore influence the evoked firing frequency. Our goal with these experiments was to see whether the presence of CNO influenced the ability of the cell to respond (in terms of action potentials generated) to incoming input, and for this reason we didn’t think it was necessary to hold the starting resting membrane potential constant (since part of the effect of CNO on AP firing may be due to its effects on baseline resting membrane potential). We did not drive the cell to achieve spike saturation. Based on the reviewer‘s concerns we have removed these data from the revised manuscript.

7) Please include the names of ImageJ plug-ins that were used for analysis. If custom java scripts were used, then please state so.

In the methods (p.26-27) we now provide details about the particular ImageJ functions that were used for the analysis.

8) sIPSC amplitudes within a cell are rarely normally distributed. Can the authors confirm that their IPSC amplitudes per cell are indeed normally distributed? If not, then please use median or cumulative probability graphs.Were sIPSCs measured from the same cells as Le-IPSCs? Activating networks using light beforehand will mean that interneuron networks will have been perturbed for spontaneous IPSC measurement.

The amplitudes of the light-evoked post-synaptic currents (le-PSCs) and spontaneous inhibitory post-synaptic currents (sIPSCs) were normally distributed based on the Anderson-Darling and Kolmogorov-Smirnov tests. Therefore, the use of unpaired t-tests to analyze the data was justified.

sIPSC and LeIPSCs were measured in the same cells but the sIPSC measurements were always taken first. Because several cells might be recorded in the same tissue slice, some of the time sIPSCs were measured after the slice had been pulsed with light used for measurements for a previously recorded cell. To determine if this might have affected sIPSC frequency, we sorted cells into those that were not exposed to light prior to recording sIPSCs and those that were and used a 2-way ANOVA to assess the effects of treatment (Dev Inhibition or Control) and light exposure. We found that treatment had an effect across both light-exposed and un-exposed cells (main effect of treatment F(1,25)=6.543, p=0.017) but light did not (no main effect of light F(1,25)=0.005194, p=0.9431) and there was no interaction of light and treatment (F(1,25)=0.4377, p=0.5143). We also found that light did not significantly change sIPSC frequency in either treatment group (post-hoc Bonferroni, p>0.9999 for both Dev Inhibition and Control groups).

9) What happens to mIPSCs? Do the authors see a change in mini amplitudes consistent with a reduction of presynaptic quantal release?

We did not measure mIPSCs so we cannot comment on this.

10) Since the authors do not see any change in PV neuron / presynaptic terminal staining, the rationale for including the results in Figure 5 is not clear. In addition, in Figure 5H, for the non-parametric KS, control has a third of the cells as dev inhib. The number of cells for these tests should be roughly equal.

Using a high magnification analysis, as suggested by reviewer 1 we now found that the total number of GAD65 puncta as well as the intensity and size of the puncta were increased in our Dev Inhibition mice. We have moved the histological data to the supplementary figures and added a short paragraph to the discuss the implications of the new findings (p. 16-17). In response to the reviewer’s concerns about the unequal numbers of cells we randomly sampled from our Dev Inhibition group to obtain the same number of cells as in the Dev Control group and found a similar distribution to what was reported in the manuscript (see Author response image 1). Analysis on this down- sampled dataset using the KS test gives a p-value of 0.11. As the distribution of the datasets appears very similar between the original and the down-sampled dataset, we have chosen to leave the original dataset that includes all the values in the manuscript.

**Author response image 1. sa2fig1:** 

For images in Figure 5B/F what were the conditions? Please show sample images for all conditions.

We now show sample images from both Control and Dev Inhibition mice.

References

1) Mukherjee, A., Carvalho, F., Eliez, S. & Caroni, P. Long-Lasting Rescue of Network and Cognitive Dysfunction in a Genetic Schizophrenia Model. *Cell* 178, 1387-1402 e1314, doi:10.1016/j.cell.2019.07.023 (2019).

2) Benoit, L. J. *et al.* Adolescent thalamic inhibition leads to long-lasting impairments in prefrontal cortex function. *Nat Neurosci* 25, 714-725, doi:10.1038/s41593-022-01072-y (2022).

3) Kozorovitskiy, Y., Saunders, A., Johnson, C. A., Lowell, B. B. & Sabatini, B. L. Recurrent network activity drives striatal synaptogenesis. *Nature* 485, 646-650, doi:10.1038/nature11052 (2012).

4) Fu, Y., Wu, X., Lu, J. & Huang, Z. J. Presynaptic GABA(B) Receptor Regulates Activity-Dependent Maturation and Patterning of Inhibitory Synapses through Dynamic Allocation of Synaptic Vesicles. *Front Cell Neurosci* 6, 57, doi:10.3389/fncel.2012.00057 (2012).

5) Wu, X. *et al.* GABA signaling promotes synapse elimination and axon pruning in developing cortical inhibitory interneurons. *J Neurosci* 32, 331-343, doi:10.1523/JNEUROSCI.3189-11.2012 (2012).

6) Baho, E. & Di Cristo, G. Neural activity and neurotransmission regulate the maturation of the innervation field of cortical GABAergic interneurons in an age-dependent manner. *J Neurosci* 32, 911-918, doi:10.1523/JNEUROSCI.4352-11.2012 (2012).

7) Chattopadhyaya, B. *et al.* GAD67-mediated GABA synthesis and signaling regulate inhibitory synaptic innervation in the visual cortex. *Neuron* 54, 889-903, doi:10.1016/j.neuron.2007.05.015 (2007).

8) Roth, B. L. DREADDs for Neuroscientists. *Neuron* 89, 683-694, doi:10.1016/j.neuron.2016.01.040 (2016).

9) Bitzenhofer, S. H., Pöpplau, J. A., Chini, M., Marquardt, A. & Hanganu-Opatz, I. L. A transient developmental increase in prefrontal activity alters network maturation and causes cognitive dysfunction in adult mice. *Neuron*, doi:10.1016/j.neuron.2021.02.011 (2021).

10) Kobayashi, Y., Ye, Z. & Hensch, T. K. Clock genes control cortical critical period timing. *Neuron* 86, 264-275, doi:10.1016/j.neuron.2015.02.036 (2015).